# Neurogliaform cortical interneurons derive from cells in the preoptic area

Mathieu Niquille[1,2†], Greta Limoni[1,2†], Foivos Markopoulos[2†], Christelle Cadilhac[1,2], Julien Prados[1,2], Anthony Holtmaat[2], Alexandre Dayer[1,2*]

[1]Department of Psychiatry, University of Geneva, Geneva, Switzerland; [2]Department of Basic Neuroscience, University of Geneva, Geneva, Switzerland

**Abstract** Delineating the basic cellular components of cortical inhibitory circuits remains a fundamental issue in order to understand their specific contributions to microcircuit function. It is still unclear how current classifications of cortical interneuron subtypes relate to biological processes such as their developmental specification. Here we identified the developmental trajectory of neurogliaform cells (NGCs), the main effectors of a powerful inhibitory motif recruited by long-range connections. Using in vivo genetic lineage-tracing in mice, we report that NGCs originate from a specific pool of 5-HT$_{3A}$R-expressing *Hmx3+* cells located in the preoptic area (POA). *Hmx3*-derived 5-HT$_{3A}$R+ cortical interneurons (INs) expressed the transcription factors PROX1, NR2F2, the marker reelin but not VIP and exhibited the molecular, morphological and electrophysiological profile of NGCs. Overall, these results indicate that NGCs are a distinct class of INs with a unique developmental trajectory and open the possibility to study their specific functional contribution to cortical inhibitory microcircuit motifs.
DOI: https://doi.org/10.7554/eLife.32017.001

*For correspondence:
alexandre.dayer@unige.ch

†These authors contributed equally to this work

Competing interests: The authors declare that no competing interests exist.

## Introduction

Cortical microcircuit function relies on the coordinated activity of a variety of GABAergic interneuron subtypes, which play critical roles in controlling the firing rate of glutamatergic pyramidal neurons, synchronizing network rhythms and regulating behavioral states (*Cardin et al., 2009*; *Fu et al., 2014*; *Kepecs and Fishell, 2014*; *Pfeffer et al., 2013*; *Pi et al., 2013*; *Pinto and Dan, 2015*; *Sohal et al., 2009*; *Zhang et al., 2014*). Different subtypes of cortical interneurons (INs) emerge during development and their specification arises through the complex interaction of cell-intrinsic mechanisms and cell-extrinsic cues (*Bartolini et al., 2013*; *Fishell and Rudy, 2011*; *Huang, 2014*; *Kessaris et al., 2014*). Cortical INs are generated in a variety of subpallial regions and the combinatorial expression of transcription factors (TFs) in these domains is believed to play a critical role in their fate specification (*Kessaris et al., 2014*; *Anastasiades and Butt, 2011*; *Flames et al., 2007*; *Wonders and Anderson, 2006*). The largest fraction (about 60–70%) of cortical INs is generated from NKX2.1-expressing progenitors located in the medial ganglionic eminence (MGE) (*Butt et al., 2008*; *Xu et al., 2008*) and their specification is under the control of the TFs LHX6 (*Du et al., 2008*; *Liodis et al., 2007*) and SOX6 (*Azim et al., 2009*; *Batista-Brito et al., 2009*). MGE-derived INs develop into fast-spiking parvalbumin (PV)+ basket and chandelier cells, as well as into Martinotti and multipolar somatostatin (SST)+ INs (*Butt et al., 2008*; *Xu et al., 2008*; *Du et al., 2008*; *Butt et al., 2005*; *Fogarty et al., 2007*; *Taniguchi et al., 2013*). The second largest fraction of cortical INs arises from the caudal ganglionic eminence (CGE) (*Miyoshi et al., 2010*; *Nery et al., 2002*) and expresses TFs such as PROX1, SP8 and NR2F2 (*Cai et al., 2013*; *Ma et al., 2012*; *Miyoshi et al., 2015*; *Rubin and Kessaris, 2013*). CGE-derived INs also express the ionotropic serotonin receptor 3A (5-HT$_{3A}$R) and give rise to a large diversity of INs, including reelin+ cells, vasointestinal peptide (VIP)+/calretinin+ bipolar cells and VIP+/cholecystokinin+ basket cells

**eLife digest** Our brain contains over a 100 billion nerve cells or neurons, and each of them is thought to connect to over 1,000 other neurons. Together, these cells form a complex network to convey information from our surroundings or transmit messages to designated destinations. This circuitry forms the basis of our unique cognitive abilities.

In the cerebral cortex – the largest region of the brain – two main types of neurons can be found: projection neurons, which transfer information to other regions in the brain, and interneurons, which connect locally to different neurons and harmonize this information by inhibiting specific messages. The over 20 different types of known interneurons come in different shapes and properties and are thought to play a key role in powerful computations such as learning and memory.

Since interneurons are hard to track, it is still unclear when and how they start to form and mature as the brain of an embryo develops. For example, one type of interneurons called the neurogliaform cells, have a very distinct shape and properties. But, until now, the origin of this cell type had been unknown.

To find out how neurogliaform cells develop, Niquille, Limoni, Markopoulos et al. used a specific gene called *Hmx3* to track these cells over time. With this strategy, the shapes and properties of the cells could be analyzed. The results showed that neurogliaform cells originate from a region outside of the cerebral cortex called the preoptic area, and later travel over long distances to reach their final location. The cells reach the cortex a few days after their birth and take several weeks to mature.

These results suggest that the traits of a specific type of neuron is determined very early in life. By labeling this unique subset of interneurons, researchers will now be able to identify the specific molecular mechanisms that help the neurogliaform cells to develop. Furthermore, it will provide a new strategy to fully understand what role these cells play in processing information and guiding behavior.

DOI: https://doi.org/10.7554/eLife.32017.002

(*Miyoshi et al., 2010*; *Armstrong and Soltesz, 2012*; *Prönneke et al., 2015*; *Lee et al., 2010*; *Murthy et al., 2014*; *Vucurovic et al., 2010*). Finally, lineage-tracing experiments using *Hmx3 (Nkx5.1)*-Cre (*Gelman et al., 2009*) and *Dbx1*-Cre driver lines (*Gelman et al., 2011*) have shown that a small fraction (about 10%) of cortical INs originate from the preoptic area (POA) (*Gelman et al., 2009*; *Gelman et al., 2011*).

Among cortical INs, neurogliaform cells (NGCs) display unique characteristics. They represent the main source of 'slow' cortical inhibition by acting on metabotropic GABA$_B$ receptors (*Tamás et al., 2003*), and are thought to be key effectors of a powerful inhibitory circuit recruited by long-range connections such as interhemispheric and thalamic projections (*Craig and McBain, 2014*; *Palmer et al., 2012*; *De Marco García et al., 2015*). Whether the current description of NGCs captures an IN subtype related to a distinct developmental specification process is unclear. Here we used in vivo genetic lineage-tracing to follow the developmental origin and trajectory of NGCs. We found that they originate from a distinct pool of 5-HT$_{3A}$R-expressing *Hmx3+* cells located in the rostral POA region, ventrally to the anterior commissure. In the embryonic POA, *Htr3a*-GFP+ INs in the *Hmx3+* domain expressed CGE-enriched TFs such as PROX1 and NR2F2, but only rarely, if not, MGE-related TFs such as NKX2.1 or LHX6. In the cortex, *Hmx3*-derived *Htr3a*-GFP+ INs expressed markers of CGE-derived INs such as NPY and/or reelin, as well as CGE-enriched TFs such as SP8, NR2F2 and PROX1, but neither MGE-specific markers such as PV or SST nor TFs such as LHX6 or SOX6. Finally, single-molecule *in situ* hybridization and electrophysiological recordings followed by *post hoc* reconstructions indicated that *Hmx3*-derived *Htr3a*-GFP+ cells exhibited the molecular, electrophysiological and morphological profile of NGCs. Taken together, these results demonstrate that cortical NGCs have a precise developmental trajectory that is linked to the expression of the transcription factor (TF) *Hmx3* in a discrete embryonic subpallial region.

# Results

To determine whether the POA could contribute to *Htr3a*-GFP INs, we crossed *Htr3a*-GFP; *R26R*-tdTOM*fl/fl* mice with *Hmx3*-Cre mice, a reporter line previously shown to fate map a population of cortical INs derived from cells located in the POA (*Gelman et al., 2009*). Examination of brains from *Hmx3*-Cre::*Htr3a*-GFP; *R26R*-tdTOM*fl/fl* matings at embryonic age 14.5 (E14.5) revealed a large fraction of *Hmx3*; tdTOM+ cells co-labelled with *Htr3a*-GFP (85.2 ± 0.9%; 1675/1986 cells in the overlap zone) in a restricted region of the POA, located ventrally to the anterior commissure (*Figure 1A,C*, *Figure 1—source data 1*). Individual co-labelled *Hmx3*; tdTOM+/*Htr3a*-GFP+ cells displaying migratory profiles were observed at more caudal levels entering the CGE (*Figure 1B*), suggesting that a fraction of *Hmx3*; tdTOM+/*Htr3a*-GFP+ cells migrate from the POA into the CGE. *In situ* hybridization indicated that the vast majority (95.8 ± 0.9%; 115/120 cells) of *Hmx3*; tdTOM+/*Htr3a*-GFP+ cells located in the POA expressed the endogenous *Htr3a* mRNA, in contrast to *Hmx3*; tdTOM+cells negative for *Htr3a*-GFP (0.5 ± 0.5%; 1/158 cells) (*Figure 1D*, *Figure 1—source data 1*). In addition, a large fraction of *Hmx3*; tdTOM+/*Htr3a*-GFP+ cells in the POA expressed the TFs PROX1 (54.0 ± 2.2%; 249/463 cells) and NR2F2 (65.9 ± 1.3%; 795/1212 cells), which have previously been shown to be enriched in CGE-derived INs (*Cai et al., 2013*; *Ma et al., 2012*; *Miyoshi et al., 2015*; *Rubin and Kessaris, 2013*), but more rarely the TF NKX2.1 (15.3 ± 1%; 175/1146 cells) (*Figure 1E,F*, *Figure 1—source data 1*). Collectively, these results indicate that a fraction of *Hmx3+* cells located in the POA express the 5-HT$_3$AR and a pattern of TFs related to CGE-derived INs.

To determine whether *Hmx3*; tdTOM+/*Htr3a*-GFP+ cells in the POA eventually give rise to a specific subpopulation of cortical INs, we examined brains at various postnatal ages. From P5 to P21, *Hmx3*; tdTOM+/*Htr3a*-GFP+ INs were found distributed preferentially in superficial cortical layers and in a variety of other brain regions including in the hippocampus (*Figure 2A*, *Figure 2—figure supplement 1*). *Hmx3*; tdTOM+/*Htr3a*-GFP+ cells were rarely observed at postnatal ages in the subpallial brain regions corresponding to the embryonic POA (i.e., the preoptic nuclei) (*Figure 2—figure supplement 2*). *In situ* hybridization for *Htr3a* mRNA indicated that *Hmx3*; tdTOM+/*Htr3a*-GFP + cells expressed the *Htr3a* transcript, similarly to *Htr3a*-GFP+ cells negative for *Hmx3*; tdTOM (*Figure 2C*). About half (51.9 ± 2.1%; 863/1653 cells) of *Hmx3*-derived cells in the cortex were co-labelled with *Htr3a*-GFP+ and virtually all *Hmx3*; tdTOM+/*Htr3a*-GFP+ (96.1 ± 0.5%; 357/372 cells) were positive for the neuronal marker NeuN (*Figure 2D,E*, *Figure 2—source data 1*). In contrast, the fraction of *Hmx3*; tdTOM+ cells negative for *Htr3a*-GFP mostly did not express NeuN (3.4 ± 1.3%; 28/758 cells) (*Figure 2D,E*, *Figure 2—source data 1*), and remained relatively constant across postnatal ages (*Figure 2—figure supplement 3A*, *Figure 2—figure Supplement 3—source data 1*). These cells displayed the morphology of glial cells and expressed the astrocytic markers GFAP and S100β as well as the oligodendrocytic marker SOX10 (*Figure 2—figure supplement 3B–D*). Overall, these findings indicate that the cortical *Hmx3*-derived lineage observed in the POA differentiate into INs that are *Htr3a*-GFP+, glial cells that are *Htr3a*-GFP negative and, for a small fraction, to NeuN+ neurons negative for *Htr3a*-GFP.

A second distinct region in the POA expressing *Dbx1* was previously reported to give rise to subsets of cortical INs (*Gelman et al., 2011*). To determine whether a fraction of *Htr3a*-GFP+ INs also originate from *Dbx1*-expressing cells, we examined *Dbx1*-Cre::*Htr3a*-GFP; *R26R*-tdTOM*fl/fl* brains at postnatal periods. While the overall contribution of *Hmx3*-derived cells to the *Htr3a*-GFP IN population in the cortex increased with postnatal maturation from P5 (6.8 ± 0.2%; 77/1118 cells) to P21 (16.0 ± 0.3%; 863/5405 cells) (*Figure 3A,C*), only minimal fractions (1.44 ± 0.2%; 81/5741 cells at P5; 0.8 ± 0.2%; 20/2551 cells at P21) of *Htr3a*-GFP+ INs were fate-mapped with *Dbx1*; tdTOM (*Figure 3B,C*, *Figure 3—source data 1*). Moreover, *Dbx1*; tdTOM+ cells were preferentially found in deep cortical layers and expressed the MGE-enriched TF SOX6 (30.4 ± 2.2%; 82/266 cells), while PROX1 was found only in a very small fraction of *Dbx1*; tdTOM+ cells expressing also the *Htr3a*-GFP (2.2 ± 0.7%; 6/266 cells) (*Figure 3D*, *Figure 3—source data 1*). In addition, *Dbx1*; tdTOM+ INs expressed *Lhx6* mRNA (*Figure 3E*), and the MGE-related markers SST and PV (*Figure 3—figure supplement 1*), and only very rarely the *Htr3a* mRNA (*Figure 3F*). Overall, these results indicate that *Hmx3*-derived 5-HT$_3$AR+ cortical INs largely originate from *Hmx3*-expressing cells but not from the *Dbx1+* domain, which gives rise to INs expressing MGE-related markers.

We next examined whether *Hmx3*; tdTOM+/*Htr3a*-GFP+ cells expressed distinct patterns of TFs involved in cortical IN subtype specification. At P21, we found that, similarly to *Htr3a*-GFP+ INs, a

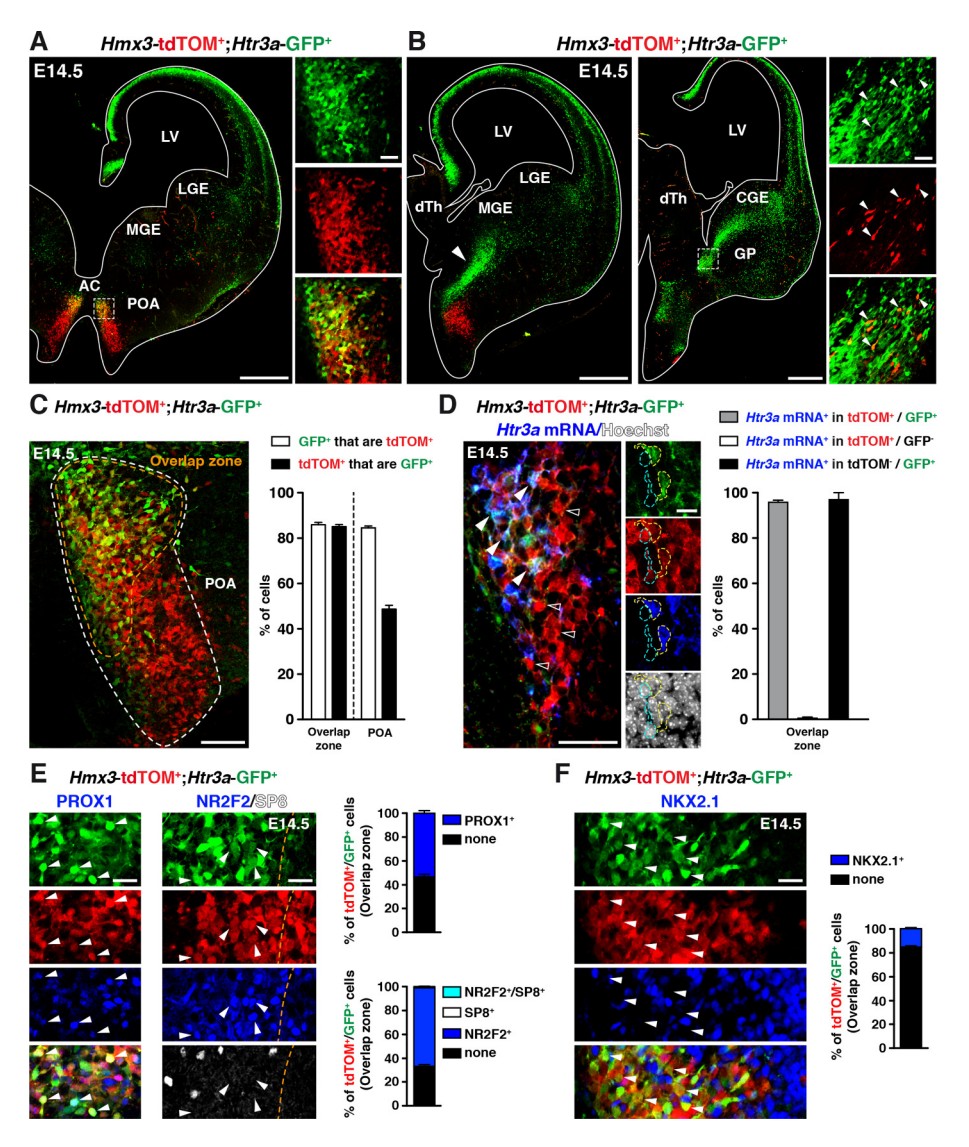

**Figure 1.** A fraction of 5-HT$_{3A}$R-expressing interneurons (INs) originates from *Hmx3*-derived cells in the preoptic area (POA) and expresses transcription factors related to the caudal ganglionic eminence (CGE). (**A**) At E14.5, tdTOM specifically labels cells expressing *Hmx3* (*Hmx3*; tdTOM+). *Htr3a*-GFP+ INs co-label with tdTOM in a rostral region of the POA (dashed lines; high magnified images) located ventrally to the anterior commissure (AC). (**B**) At more caudal levels, *Hmx3*; tdTOM+/*Htr3a*-GFP+ embryonic cells appear to further migrate caudally (arrowhead) towards the CGE. High magnified images show *Hmx3*; tdTOM+/*Htr3a*-GFP+ cells entering the ventral CGE (dashed lines). (**C**) More than 80% of *Hmx3*; tdTOM+ cells co-label with *Htr3a*-GFP (and conversely) in the overlap zone (orange dashed line) of the POA domain defined by *Hmx3*; tdTOM recombination (white dashed line). (**D**) *In situ* hybridization showing that almost all *Hmx3*; tdTOM+/*Htr3a*-GFP+ INs in the POA express the *Htr3a* mRNA (arrowheads; yellow outline), whereas *Hmx3*; tdTOM+ cells do not (empty arrowheads; cyan outline). (**E**) In the overlap zone of the POA, IHC reveals that *Hmx3*; tdTOM+/*Htr3a*-GFP+ embryonic cells express (arrowheads) the CGE-enriched transcription factors PROX1 (left) and NR2F2 but not SP8 (right). (**F**) By contrast, the vast majority of *Hmx3*; tdTOM+/*Htr3a*-GFP+ INs do not express NKX2.1 (arrowheads). dTh: dorsal thalamus, GP: globus pallidus, LGE: lateral ganglionic eminence, LV: lateral ventricle, MGE: medial ganglionic eminence. Scale bars: 300 μm in A, B: low magnified images; 100 μm in C, D: low magnified images; 50 μm in A, B: high magnified images; 25 μm in E, F; 10 μm in D: high magnified images.

DOI: https://doi.org/10.7554/eLife.32017.003

The following source data is available for figure 1:

**Source data 1.** Detailed counts of cells quantified in *Figure 1* in the different experimental conditions.

DOI: https://doi.org/10.7554/eLife.32017.004

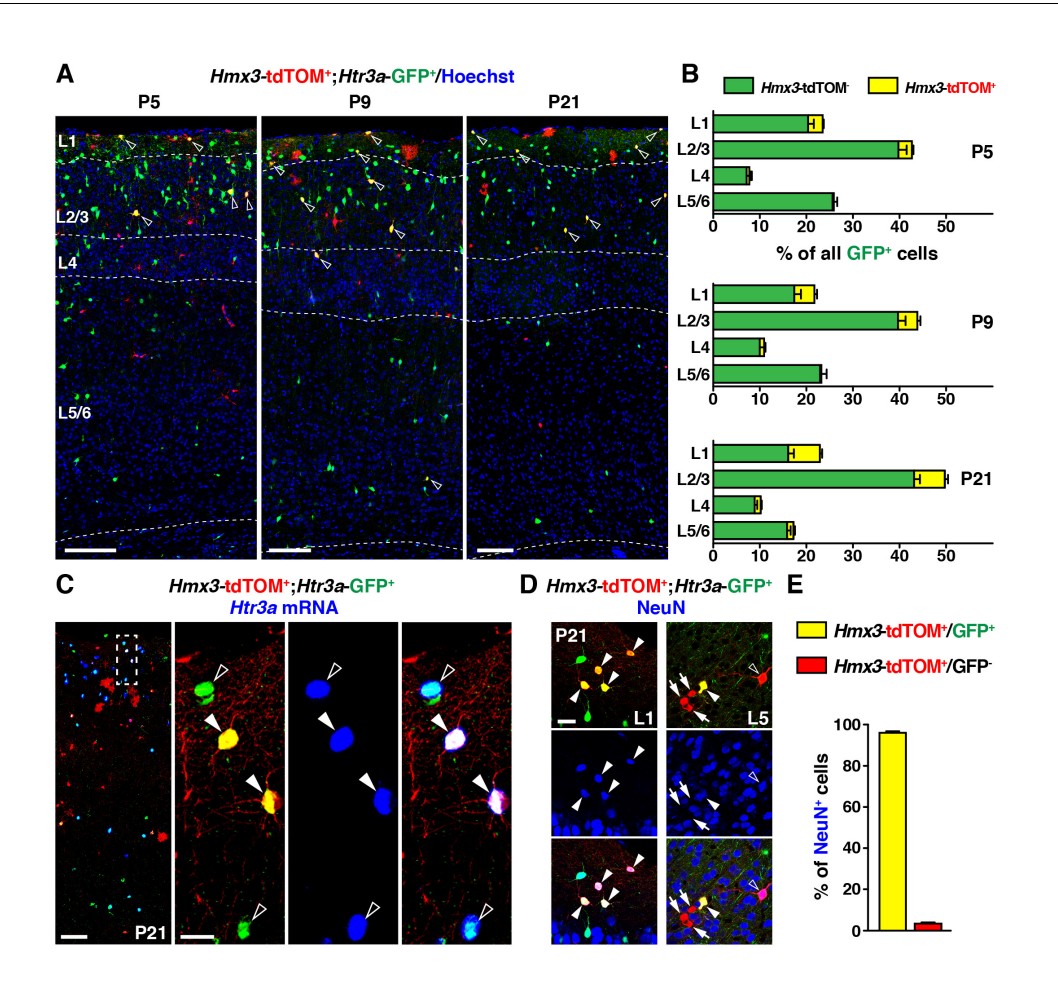

**Figure 2.** During the postnatal period, *Hmx3*-derived cells from the preoptic area (*Gelman et al., 2009*) constitute a small but persistent fraction of 5-HT$_{3A}$R-expressing interneurons (INs) (*Lee et al., 2010*) in superficial cortical layers. (**A–B**) *Hmx3*; tdTOM+/*Htr3a*-GFP+ cells represent a fraction of *Htr3a*-GFP+ INs that increases along postnatal ages P5 (left), P9 (middle) and P21 (right). Note that double-labeled cells (open arrowheads) are mainly located in superficial cortical layers. (**C**) *In situ* hybridization showing that, at P21, *Htr3a* mRNA is found in cortical *Hmx3*; tdTOM+/*Htr3a*-GFP+ INs (arrowheads) as in *Htr3a*-GFP+ INs (open arrowheads). (**D–E**) *Hmx3*; tdTOM+/*Htr3a*-GFP+ cells largely express the neuronal marker NeuN (arrowheads) whereas those negative for *Htr3a*-GFP only rarely do (arrows, open arrowhead). Scale bars: 100 µm in A, C: low magnified images; 25 µm in C: high magnified images, D.

DOI: https://doi.org/10.7554/eLife.32017.005

The following source data and figure supplements are available for figure 2:

**Source data 1.** Detailed counts of cells quantified in *Figure 2* in the different experimental conditions.
DOI: https://doi.org/10.7554/eLife.32017.010

**Figure supplement 1.** Rostro-caudal distribution of *Hmx3*; tdTOM+/*Htr3a*-GFP+ cells at P21.
DOI: https://doi.org/10.7554/eLife.32017.006

**Figure supplement 2.** At postnatal ages, *Hmx3*; tdTOM+/*Htr3a*-GFP+ cells are rarely observed in the preoptic nuclei (PoN), the subpallial brain region corresponding to the embryonic preoptic area.
DOI: https://doi.org/10.7554/eLife.32017.007

**Figure supplement 3.** *Hmx3*; tdTOM+ cells negative for *Htr3a*-GFP often express glial markers.
DOI: https://doi.org/10.7554/eLife.32017.008

**Figure supplement 3—source data 1.** Detailed counts of cells quantified in *Figure 2—figure supplement 3* in the different experimental conditions.
DOI: https://doi.org/10.7554/eLife.32017.009

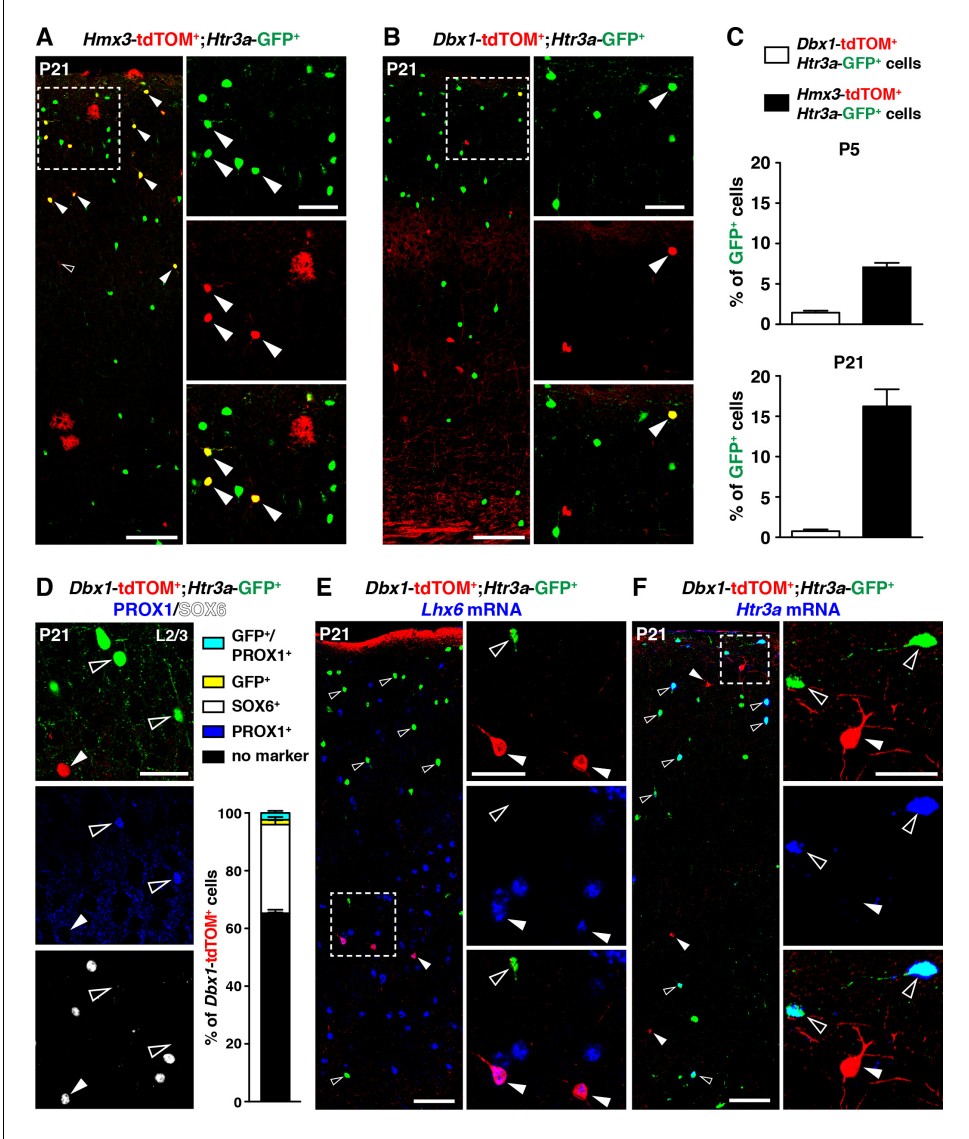

**Figure 3.** 5-HT$_{3A}$R-expressing interneurons (INs) largely originate from *Hmx3*+ but not *Dbx1*+ cells. (**A–C**) A consistent fraction of cortical *Htr3a*-GFP+ INs co-labels with *Hmx3*; tdTOM (A, arrowheads) at P5 and P21, whereas only a minimal fraction does with *Dbx1*; tdTOM (B, arrowhead). (**D**) *Dbx1*; tdTOM+ INs express the MGE-enriched TF SOX6 (arrowhead) but not the CGE-enriched TF PROX1 (open arrowheads). Only a minimal fraction of *Dbx1*; tdTOM+ co-labelled for *Htr3a*-GFP, among which the majority were PROX1+. (**E–F**) *In situ* hybridization showing that *Dbx1*; tdTOM+ INs express the transcript for the MGE-enriched TF *Lhx6* (E, arrowheads) whereas *Htr3a*-GFP + INs do not (E, open arrowheads). In contrast, *Dbx1*; tdTOM+ INs do not express the *Htr3a* transcript (F, arrowheads) whereas *Htr3a*-GFP+ INs do (F, open arrowheads). Scale bars: 100 μm in A, B, E, F: low magnified images; 50 μm in A, B, D, E, F: high magnified images.

DOI: https://doi.org/10.7554/eLife.32017.011

The following source data and figure supplement are available for figure 3:

**Source data 1.** Detailed counts of cells quantified in *Figure 3* in the different experimental conditions.
DOI: https://doi.org/10.7554/eLife.32017.013
**Figure supplement 1.** Interneurons (INs) derived from *Dbx.1*+ cells express MGE-enriched markers.
DOI: https://doi.org/10.7554/eLife.32017.012

large fraction (65.8 ± 3.4%; 202/308 cells) of *Hmx3*; tdTOM+/*Htr3a*-GFP+ INs expressed the CGE-enriched but not the MGE-related TFs. Indeed, a large fraction of them (65.8 ± 3.4%; 202/308 cells) expressed PROX1_but not SOX6 (*Figure 4A,B*, *Figure 4—source data 1*), as well as NR2F2 (32.7 ±

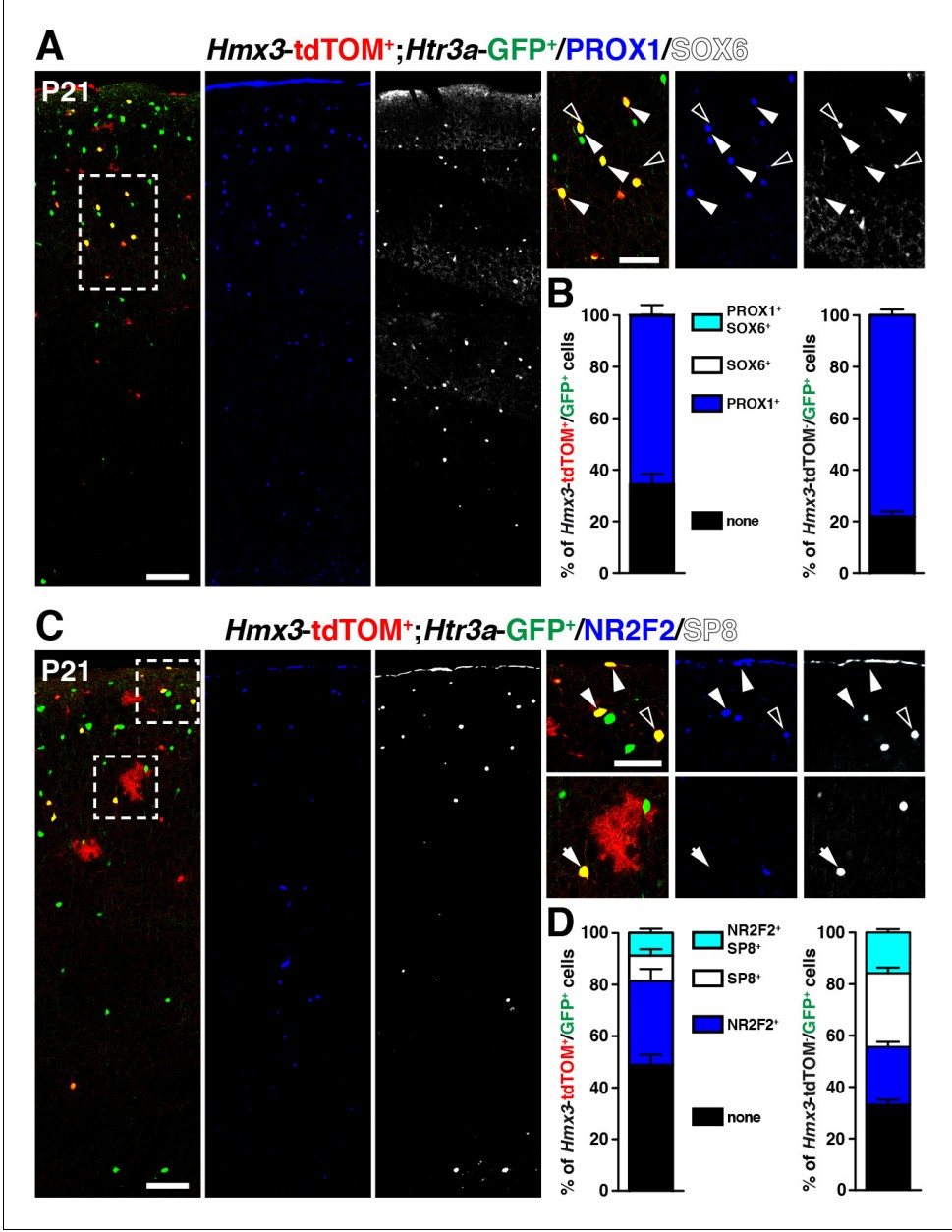

**Figure 4.** *Hmx3*; tdTOM+/*Htr3a*-GFP+ cortical interneurons (INs) express markers related to the CGE but not to the MGE. (**A–B**) *Hmx3*; tdTOM+/*Htr3a*-GFP+ INs express the CGE-enriched TF PROX1 (A; arrowheads) but not the MGE-related TF SOX6 (A; open arrowheads) similarly to *Htr3a*-GFP+ INs that do not derive from *Hmx3*+ cells (B, right graph). (**C–D**) *Hmx3*; tdTOM+/*Htr3a*-GFP+ cells express the CGE-enriched TFs NR2F2 (arrowheads) and (open arrowheads)/or SP8 (arrow). *Htr3a*-GFP+ derived from *Hmx3*+ cells are biased towards NR2F2 expression, in comparison to *Htr3a*-GFP+ INs that do not co-label with *Hmx3*; tdTOM (D, blue bars). Scale bars: 100 µm in A, C: low magnified images; 50 µm in A, C: high magnified images.

DOI: https://doi.org/10.7554/eLife.32017.014

The following source data and figure supplement are available for figure 4:

**Source data 1.** Detailed counts of cells quantified in *Figure 4* in the different experimental conditions.
DOI: https://doi.org/10.7554/eLife.32017.016

**Figure supplement 1.** *Hmx3*; tdTOM+/*Htr3a*-GFP+ cortical interneurons (INs) display a combinatorial expression of NR2F2 and SP8 that is biased toward NR2F2.
DOI: https://doi.org/10.7554/eLife.32017.015

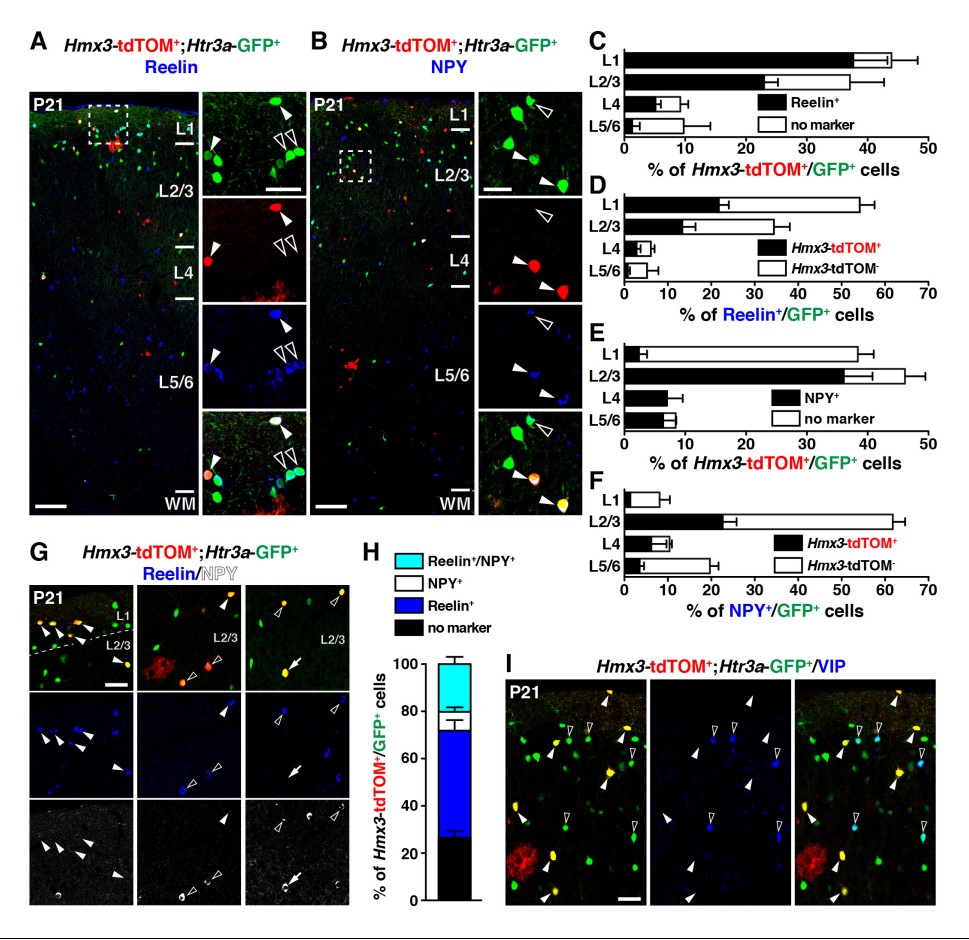

**Figure 5.** *Hmx3*; tdTOM+/*Htr3a*-GFP+ cortical interneurons (INs) express reelin and NPY but not VIP. (A–F) *Hmx3*; tdTOM+/*Htr3a*-GFP+ INs are stained with the neurochemical markers reelin (A, arrowheads) and NPY (B, arrowheads) as well as *Htr3a*-GFP+ cells negative for *Hmx3*; tdTOM (open arrowheads). Note that reelin-positive *Hmx3*; tdTOM+/*Htr3a*-GFP+ INs are preferentially found in L1-3 (C) whereas NPY-expressing *Hmx3*; tdTOM+/ *Htr3a*-GFP+ INs are mainly found in L2-6 (E). *Hmx3*; tdTOM+/*Htr3a*-GFP+ INs account for more than one third of all reelin-positive *Htr3a*-GFP+ (D) and of all NPY-positive *Htr3a*-GFP+ cells (F). (G–H) *Hmx3*; tdTOM+/*Htr3a*-GFP + INs mainly express reelin (G, arrowheads) or reelin and NPY (G, open arrowheads) but to a smaller extend only NPY (G, arrow). (I) *Hmx3*; tdTOM+/*Htr3a*-GFP+ INs do not express VIP (arrowheads), whereas *Htr3a*-GFP+ INs negative for *Hmx3*; tdTOM do (open arrowheads). WM: white matter. Scale bars: 100 µm in A, B: low magnified images; 50 µm in A, B: high magnified images, G, I.
DOI: https://doi.org/10.7554/eLife.32017.017

The following source data and figure supplement are available for figure 5:

**Source data 1.** Detailed counts of cells quantified in *Figure 5* in the different experimental conditions.
DOI: https://doi.org/10.7554/eLife.32017.019

**Figure supplement 1.** *Hmx3*; tdTOM+/*Htr3a*-GFP+ do not express MGE-enriched markers.
DOI: https://doi.org/10.7554/eLife.32017.018

5.9%; 71/218 cells), SP8 (9.8 ± 2.6%; 22/218 cells), and both SP8 and NR2F2 (8.8 ± 2.0%; 18/218 cells) (*Figure 4C,D*, *Figure 4—source data 1*). The fraction of *Hmx3*; tdTOM+/*Htr3a*-GFP+ expressing at least one of these two latter TFs was smaller and biased toward NR2F2 expression, when compared to *Htr3a*-GFP+ INs (*Figure 4D*, *Figure 4—figure supplement 1*, *Figure 4—source data 1*). These findings indicate that *Hmx3*; tdTOM+/*Htr3a*-GFP+ cortical INs express a repertoire of TFs related to CGE but not to MGE-derived INs.

We next examined whether *Hmx3*; tdTOM+/*Htr3a*-GFP+ INs expressed classical CGE markers such as reelin, NPY and VIP (*Lee et al., 2010*; *Murthy et al., 2014*; *Vucurovic et al., 2010*).

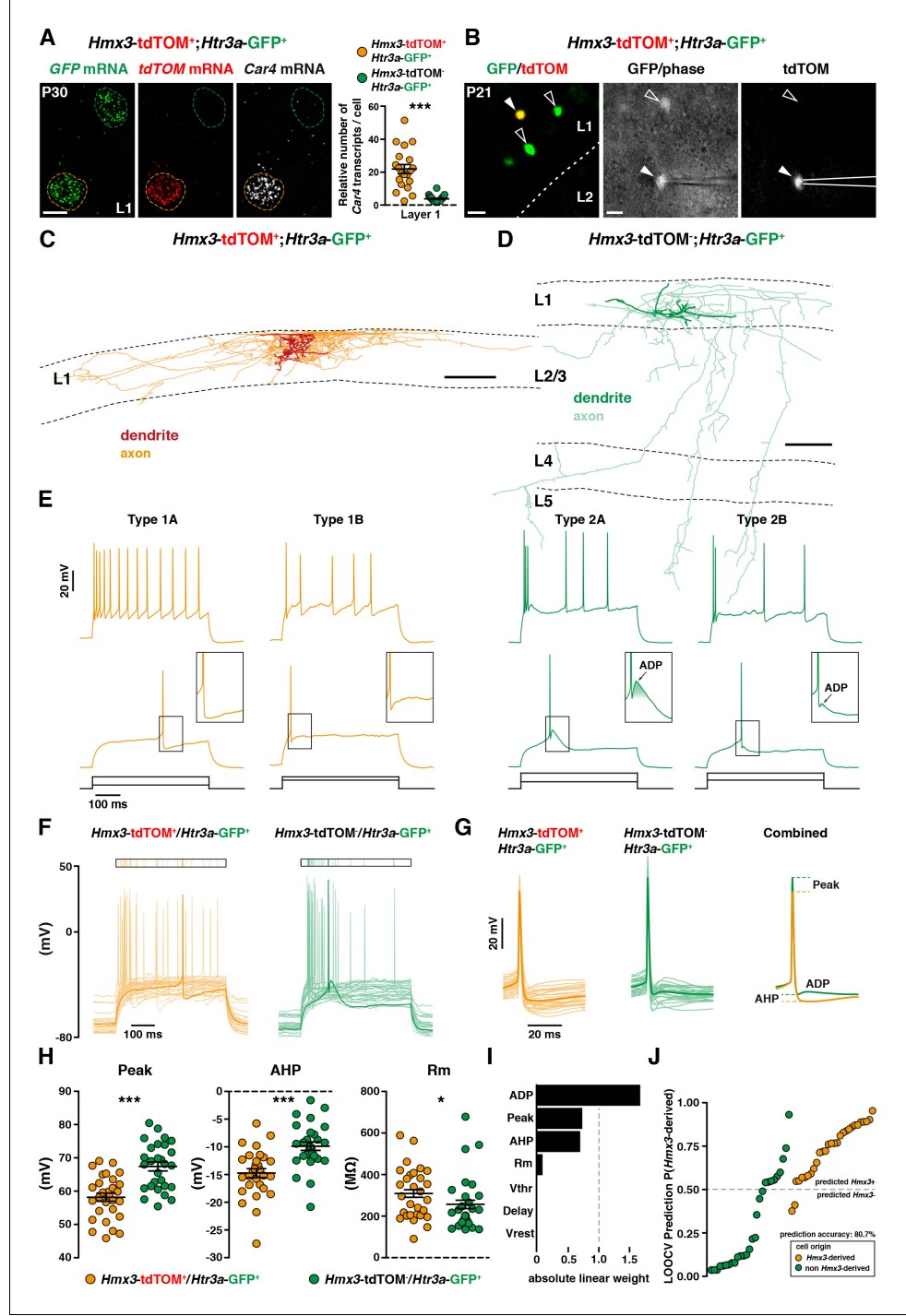

**Figure 6.** *Hmx3*; tdTOM+/*Htr3a*-GFP+ cortical interneurons (INs) in layer 1(L1) display the molecular, morphological and electrophysiological features of neurogliaform cells (NGCs). (**A**) RNAscope multiplex fluorescent hybridization for *tdTOM*, *GFP* and *Car4* transcripts on P30 brains showing that L1 *Hmx3*; tdTOM+/ *Htr3a*-GFP+ INs express *Car4* at significantly higher levels (orange outline) as compared to *Htr3a*-GFP+ INs negative for tdTOM (green outline) (***p<0.0001; Mann-Whitney test). (**B**) Example of a L1 *Hmx3*; tdTOM+/*Htr3a*-GFP+ cell (arrowhead) or *Htr3a*-GFP+ INs negative for tdTOM (open arrowheads) checked before patching (left image). Example of a patched *Hmx3*; tdTOM+/*Htr3a*-GFP+ IN (middle and right images, arrowhead). (**C**) Illustrative reconstruction of a *Hmx3*; tdTOM+/*Htr3a*-GFP+ IN in L1 displaying the characteristic morphology of an elongated NGC with dense axonal ramifications restricted to L1. (**D**) Illustrative reconstruction of a *Htr3a*-GFP+ IN negative for tdTOM in L1 displaying the characteristic morphology of single bouquet-like cell (SBC) with axonal

*Figure 6 continued on next page*

*Figure 6 continued*

ramifications extending deep into L5. (**E**) Illustrative traces from recorded *Hmx3*; tdTOM+/*Htr3a*-GFP+ INs (orange) and *Htr3a*-GFP+ INs negative for tdTOM (green), showing the first action potentials (APs) at rheobase and trains of APs at higher current injections. (**F**) Superimposed single AP traces at rheobase of all *Hmx3*; tdTOM +/*Htr3a*-GFP+ INs (orange) and *Htr3a*-GFP+ INs negative for tdTOM (green). Thick traces correspond to type 1A and type 2A examples in E. (**G**) Same traces as in (**F**), aligned to the AP, with a lower time scale. Thin lines are individual cell traces and thick lines are trace averages. The average traces on the right are aligned to the threshold potential (Vthr). (**H**) Plots of AP peak amplitude (Peak; ***p<0.0001; unpaired t-test), after hyperpolarization potential amplitude (AHP; ***p<0.0001; unpaired t-test) and membrane resistance (Rm; *p=0.0318; Mann-Whitney test) showing significant differences between the two cell types. (**I**) Absolute linear weights assigned by the classification model trained on all cells with standardized electrophysiological properties. (**J**) Prediction probabilities estimated by the classifier on the cell left out in the leave-one-out-cross-validation (LOOCV) loop. Cells are ordered on the x-axis by origin and prediction value, and the color code reflect their origin. Cells above the probability threshold 0.5 are more likely to be *Hmx3*-derived according to the model. Scale bars: 10 µm A; 20 µm in B; 100 µm C, D.

DOI: https://doi.org/10.7554/eLife.32017.020

The following source data and figure supplements are available for figure 6:

**Source data 1.** Detailed counts of cells expressing *Car4* quantified in *Figure 6* in the different experimental conditions.

DOI: https://doi.org/10.7554/eLife.32017.025

**Source data 2.** Electrophysiological properties of cells quantified in *Figure 6*.

DOI: https://doi.org/10.7554/eLife.32017.026

**Figure supplement 1.** *Hmx3*; tdTOM+/*Htr3a*-GFP+ interneurons (INs) in layer 1 (L1) display the characteristic morphology of elongated neurogliaform cells (eNGCs).

DOI: https://doi.org/10.7554/eLife.32017.021

**Figure supplement 2.** Prediction model parameters and electrophysiological features of *Hmx3*; tdTOM+/*Htr3a*-GFP+ interneurons (INs) in layer 1.

DOI: https://doi.org/10.7554/eLife.32017.022

**Figure supplement 3.** *Hmx3*; tdTOM+/*Htr3a*-GFP+ interneurons (INs) in cortical layers 2–6 (L2-6) display molecular and electrophysiological properties of NGCs.

DOI: https://doi.org/10.7554/eLife.32017.023

**Figure Supplement 3—source data 1.** Detailed counts of cells expressing *Car4* quantified in *Figure 6—figure supplement 3* in the different experimental conditions.

DOI: https://doi.org/10.7554/eLife.32017.027

**Figure Supplement 3—source data 2.** Electrophysiological properties of cells quantified in *Figure 6—figure supplement 3*.

DOI: https://doi.org/10.7554/eLife.32017.028

**Figure supplement 4.** *Hmx3*; tdTOM+/*Htr3a*-GFP+ interneurons (INs) in cortical layers 2–6 (L2-6) display the characteristic morphology of NGCs.

DOI: https://doi.org/10.7554/eLife.32017.024

Quantification across layers revealed that a large fraction of *Hmx3*; tdTOM+/*Htr3a*-GFP+ INs expressed reelin or NPY. This was particularly striking in layer 1 (L1) for reelin (*Figure 5A,C*) and in L2–6 for NPY (*Figure 5B,E*), respectively (*Figure 5—source data 1*). Overall, *Hmx3*; tdTOM+/*Htr3a*-GFP+ INs accounted for approximately a third of all reelin+/*Htr3a*-GFP+ INs (34.5 ± 2.3%; 267/797 cells) and of all NPY+/*Htr3a*-GFP+ INs (27.7 ± 2.3%; 149/571 cells) (*Figure 5D,F*, *Figure 5—source data 1*). Given that INs expressing reelin have been shown to co-express NPY (*Lee et al., 2010*), we assessed reelin and NPY co-expression in *Hmx3*; tdTOM+/*Htr3a*-GFP+ cells. At P21, only a small fraction (8.0 ± 0.9%; 17/232 cells) of these cells expressed NPY without reelin, thus indicating that reelin labels the largest fraction (66.1 ± 8.6%; 267/398 cells) of *Hmx3*-derived *Htr3a*-GFP+ INs (*Figure 5G,H*, *Figure 5—source data 1*). In contrast, *Hmx3*; tdTOM+/*Htr3a*-GFP+ INs did not co-label nor with the CGE-specific marker VIP (*Figure 5I*) neither with the MGE-enriched markers SST and PV (*Figure 5—figure supplement 1*). These results indicate that *Hmx3*-derived *Htr3a*-GFP+ INs mainly belong to the reelin but not to the VIP subtypes and account for an important fraction of all reelin+/*Htr3a*-GFP+ INs.

Two distinct profiles of reelin-expressing INs have been identified in L1 of the neocortex, namely neurogliaform (NGCs) and single bouquet-like cells (SBCs) (*Cadwell et al., 2016*; *Jiang et al.,*

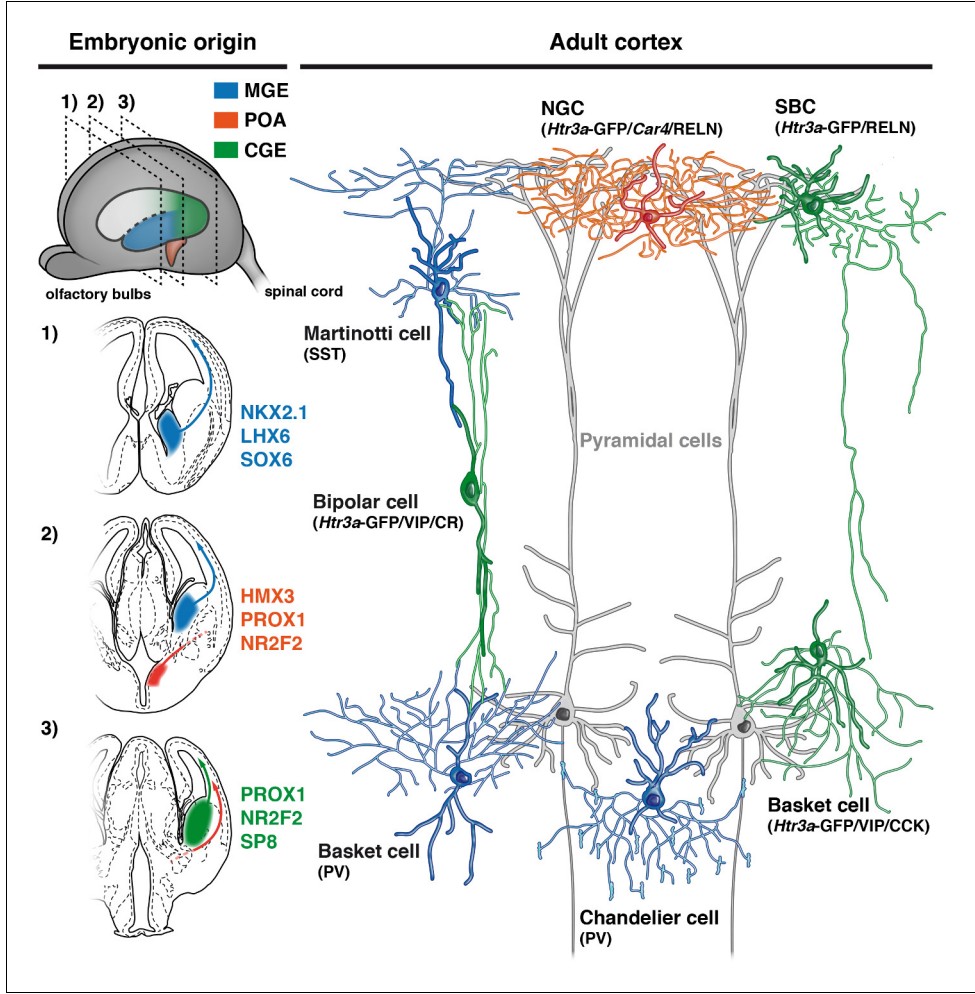

**Figure 7.** Developmental origin of cardinal classes of cortical interneurons. The Martinotti somatostatin (SST) cell, the parvalbumin (PV) basket cell and the PV+ chandelier cell originate from NKX2.1+ progenitors of the medial ganglionic eminence (MGE, blue) and rely on the transcription factors (TFs) LHX6 and SOX6. The *Htr3a*-GFP/reelin (RELN) single-bouquet cell (SBC), the *Htr3a*-GFP/vasointestinal peptide (VIP)/cholecystokinin (CCK) basket cell and the *Htr3a*-GFP/VIP/calretinin (CR) bipolar cell are derived from cells located in the caudal ganglionic eminence (CGE, green) that express the TFs PROX1, NR2F2 and SP8. The RELN/*Car4*/*Htr3a*-GFP neurogliaform cell (NGC) is specifically derived from *Hmx3*+ cells located in the preoptic area (POA, orange) and express the TFs PROX1 and NR2F2.

DOI: https://doi.org/10.7554/eLife.32017.029

2015). At a molecular level, NGCs are strongly enriched in the carbonic anhydrase 4 (*Car4*) transcript in contrast to SBCs (*Cadwell et al., 2016*). Using single-molecule fluorescent *in situ* hybridization experiments (*Wang et al., 2012*), we found that *Hmx3*; tdTOM+/*Htr3a*-GFP+ INs in L1 exhibited significantly higher levels of *Car4* transcripts in contrast to *Htr3a*-GFP+ INs (*Figure 6A*, *Figure 6— source data 1*), thus indicating that *Hmx3*-derived *Htr3a*-GFP+ INs share the molecular profile of NGC. To further verify whether their morphological and electrophysiological features could fit with NGCs, we performed whole-cell recordings (*Figure 6B*) and reconstructions (*Figure 6C,D*; *Figure 6—figure supplement 1*) of *Hmx3*-derived versus non *Hmx3*-derived *Htr3a*-GFP+ INs in L1 of the barrel cortex. There, *Hmx3*; tdTOM+/*Htr3a*-GFP+ INs displayed the characteristic morphology of elongated NGCs with dense axonal ramifications mostly restricted to L1 (*Figure 6C*, *Figure 6— figure supplement 1*), whereas *Htr3a*-GFP+ INs negative for tdTOM had the morphology of SBCs with less developed axonal processes that extended deeper in cortical layers (*Figure 6D*). With regard to their first action potential (AP) at rheobase, NGCs are reported to display a type 1 profile

with only an after-hyperpolarization potential (AHP), whereas SBCs show a type 2 profile consisting of an AHP followed by an after-depolarization potential (ADP) (*Cadwell et al., 2016*; *Jiang et al., 2015*). Strikingly, all (29 out of 29) recorded *Hmx3*; tdTOM+/*Htr3a*-GFP+ INs were of type 1, thus confirming their NGC identity. Moreover, the vast majority of them (23 out of 29) were of type 1A with a deep and wide AHP and only a few (6 out of 29) were of type 1B with a shallow and narrow AHP (*Figure 6E*, left, *Figure 6—source data 2*). *Htr3a*-GFP+ INs negative for *Hmx3*; tdTOM had more variable profiles, but the majority of them (24 out of 28) were displaying a type two profile with an average ADP amplitude of 2.30 ± 0.51 mV, suggesting that they were SBCs. Most of them (20 out of 28) were of type 2B with a small ADP below the spike threshold, a few others (4 out of 28) were of type 2A with a big ADP above the spike threshold (*Figure 6E*, right, *Figure 6—source data 2*) and another few of them (4 out of 28) had not measurable ADP (not shown). *Hmx3*; tdTOM+/*Htr3a*-GFP+ INs showed also a higher tendency to late-spiking when compared to *Htr3a*-GFP+ INs (*Figure 6F*, *Figure 6—source data 2*). *Hmx3*; tdTOM+/*Htr3a*-GFP+ INs had bigger AP delay average, but not significantly different from *Htr3a*-GFP+ INs negative for *Hmx3*; tdTOM (*Figure 6—figure supplement 2C*, *Figure 6—source data 2*). However, the variability of individual cell values was higher for *Hmx3*; tdTOM+/*Htr3a*-GFP+ INs, indicating that these cells tend to be more late-spiking, a characteristic of NGCs (*Cadwell et al., 2016*; *Jiang et al., 2015*). Alignement of the first APs at rheobase revealed other putative differences between the two groups (*Figure 6G*, *Figure 6—source data 2*). After quantification, *Hmx3*; tdTOM+/*Htr3a*-GFP+ INs significantly differed from *Htr3a*-GFP+ INs negative for *Hmx3*; tdTOM in the first AP amplitude (Peak), AHP amplitude (AHP), membrane resistance (Rm) (*Figure 6H*, *Figure 6—source data 2*) and threshold potential (Vthr) (*Figure 6—figure supplement 2C*). We next aimed to determine whether *Hmx3* and non *Hmx3*-derived *Htr3a*-GFP+ INs classes in L1 could be predicted from single-cell electrophysiological properties. Using an automatic cell type classifier based on combined electrophysiological measures, we were able to predict the *Hmx3*-derived class with 80.7% accuracy, with highest weights found on ADP, Peak and AHP but not Vthr (*Figure 6I,J*; *Figure 6—figure supplement 2A,B*, *Figure 6—source data 2*). Finally, we analysed *Hmx3*; tdTOM+/*Htr3a*-GFP+ INs in other cortical layers to determine whether they displayed the same NGC characteristics. Similarly to L1 cells, *Car4* expression in L2-6 was significantly higher in *Hmx3*; tdTOM+/*Htr3a*-GFP+ INs as compared to *Htr3a*-GFP+ INs negative for tdTOM (*Figure 6—figure supplement 3A,B*, *Figure 6—source data 1*). Morphological recovery of *Hmx3*; tdTOM+/*Htr3a*-GFP+ INs located in L2–6 revealed that all cells (7 out of 7) had also the characteristic morphology of NGCs (*Figure 6—figure supplement 4*). Furthermore, characteristic properties of NGC like the tendency to late spiking, the depth of AHP, and the level of Vthr were significantly more pronounced in these cells compared to L1 cells (*Figure 6—figure supplement 3C,E*). Overall, these data indicate that *Htr3a*-GFP+ INs displaying the molecular, morphological and electrophysiological properties of NGC INs originate from *Hmx3*-expressing cells in the embryonic POA (*Figure 7*, orange), whereas SBCs in layer 1, as well as VIP +INs, are more likely to originate from the CGE (*Figure 7*, green).

## Discussion

Distinct subtypes of local GABAergic INs are required to regulate microcircuit function (*Cardin et al., 2009*; *Fu et al., 2014*; *Kepecs and Fishell, 2014*; *Pfeffer et al., 2013*; *Pi et al., 2013*; *Pinto and Dan, 2015*; *Sohal et al., 2009*; *Zhang et al., 2014*). Whether current classifications of cortical IN subtypes relate to intrinsic biological processes such as their developmental specification is a key question in the field (*Huang, 2014*; *Taniguchi et al., 2013*). Among cortical INs, NGCs are considered as belonging to a distinct subtype acting as a main effector of a powerful inhibitory motif recruited by long-range connections (*Tamás et al., 2003*; *Craig and McBain, 2014*; *Palmer et al., 2012*). Here we aimed to track the developmental trajectory of NGCs using genetic fate mapping strategies. We find that NGCs derive from *Htr3a*-GFP+/*Hmx3*+ cells located in the embryonic POA, but not from *Dbx1*+ cells. Strikingly, L1 *Hmx3*-derived *Htr3a*-GFP+ INs display the distinct molecular, morphological and electrophysiological properties of NGCs, whereas *Htr3a*-GFP+ INs negative for *Hmx3* have the profile of SBCs (*Cadwell et al., 2016*). *Hmx3*-derived *Htr3a*-GFP+ NGCs represent about a third of reelin-expressing *Htr3a*-GFP+ INs. At a molecular level, *Hmx3*; tdTOM+/*Htr3a*-GFP+ NGCs express CGE-related TFs such as NR2F2 and PROX1, indicating that they share common features with CGE-derived INs. Overall, these results indicate that cortical NGCs

derive from a discrete embryonic area located in the subpallial POA and that their specification is linked to the expression of the TF *Hmx3*.

## *Htr3a*-GFP+ INs derived from *Hmx3+* cells express CGE-enriched transcription factors

Here, we find that a fraction (about 15%) of *Htr3a*-GFP+ cortical INs originate from *Hmx3+* but not *Dbx1+* cells in the POA. The overall fraction of *Hmx3*; tdTOM+/*Htr3a*-GFP+ INs to the total *Htr3a*-GFP+ IN population in the cortex almost doubled from P9 to P21, a period during which neural migration is largely achieved. Given that about 40% of developing cortical INs undergo apoptosis during early postnatal life (*Southwell et al., 2012*) higher levels of programmed cell death in *Htr3a*-GFP+ INs negative for tdTOM could thus account for the relative postnatal increase in the cortical *Hmx3*; tdTOM+/*Htr3a*-GFP+ cell population. Overall, our data support the general view that the differential expression of TFs in progenitor cells originating from distinct subpallial germinal zones controls the specification of cortical IN subtypes (*Huang, 2014*; *Kessaris et al., 2014*; *Anastasiades and Butt, 2011*; *Flames et al., 2007*; *Gelman et al., 2009*). A striking example in the field relates to chandelier INs, which have been shown to derive from *Nkx2.1+* cells produced specifically at late embryonic time-points in a restricted region of the MGE (*Taniguchi et al., 2013*). Three major germinal zones contribute to the generation of cortical INs, including the MGE, the CGE and the POA (*Kessaris et al., 2014*). The majority of cortical IN subtypes (about 60–70%) originates from *Nkx2.1+* progenitors in the MGE and includes fast-spiking PV+ basket INs, chandelier cells and SST + Martinotti cells. In addition to NKX2.1, sequential expression of the TFs such as LHX6 (*Anastasiades and Butt, 2011*; *Du et al., 2008*; *Liodis et al., 2007*) and SOX6 (*Azim et al., 2009*; *Batista-Brito et al., 2009*) controls the specification and migration of MGE-derived IN subtypes. Here we find that *Htr3a*-GFP+ cortical INs originating from *Hmx3+* cells in the POA do not express MGE-enriched TFs such as LHX6 or SOX6. In the embryonic POA, we observe that only a small fraction of *Hmx3*; tdTOM+/*Htr3a*-GFP+ cells expresses the TF NKX2.1, which has been shown to be strongly expressed in the ventricular zone of the POA (*Flames et al., 2007*). This could be due to either down-regulation of NKX2.1 in postmitotic *Hmx3*; tdTOM+/*Htr3a*-GFP+ cells as previously observed in migrating MGE-derived INs (*Nóbrega-Pereira et al., 2008*) or to the fact that the majority of *Hmx3*; tdTOM+/*Htr3a*-GFP+ cells do not originate from NKX2.1 progenitors. In line with this second possibility, recent genetic fate-mapping experiments using a *Nkx2.1*-ires-Flpo knock-in mouse line did not appear to label INs in L1 (*He et al., 2016*). Overall, further work needs to be done to clarify the precise origin of mitotic cells giving rise to the pool of *Hmx3*; tdTOM+/*Htr3a*-GFP+ cells observed in the embryonic POA. In contrast to the absence of co-localization with MGE-enriched TFs, we find that *Hmx3*; tdTOM+/*Htr3a*-GFP+ INs express TFs such as PROX1 and NR2F2 in the embryonic POA and in the postnatal cortex. PROX1 and NR2F2 have been shown to be expressed in CGE cells and these TFs are maintained in subsets of cortical INs as they mature in the developing cortex (*Cai et al., 2013*; *Ma et al., 2012*; *Rubin and Kessaris, 2013*; *Murthy et al., 2014*; *Kanatani et al., 2008*). Our results thus indicate that the specification of *Hmx3*-derived and CGE-derived *Htr3a*-GFP+ INs shares common transcriptional controls and that the expression of *Hmx3* in a fraction of *Htr3a*-GFP+ defines the distinct subtype of NGCs. To gain insights on the requirement of *Hmx3* in the specification of *Hmx3*-derived *Htr3a*-GFP+ NGCs, cell-type specific genetic deletion strategies are needed. Finally, the molecular pathways specifically controlled by *Hmx3* in NGCs remain to be identified.

## *Htr3a*-GFP+ INs derived from *Hmx3+* cells express CGE but not MGE-enriched neurochemical markers

MGE-derived INs express the neurochemical markers PV or SST and are preferentially distributed in lower cortical layers, whereas CGE-derived INs specifically express the 5-HT$_{3A}$R, but not PV or SST, and populate more superficial cortical layers (*Fishell and Rudy, 2011*; *Huang, 2014*; *Rudy et al., 2011*). Using *in situ* hybridization, we confirmed that *Hmx3+* lineage give rise to superficial cortical *Htr3a*-GFP+ INs expressing the *Htr3a* transcript. Reelin, VIP and NPY have been used as neurochemical markers to identify different subtypes of *Htr3a*-GFP+ cortical INs (*Lee et al., 2010*; *Murthy et al., 2014*; *Vucurovic et al., 2010*). Expressions of reelin and VIP are mutually exclusive in *Htr3a*-GFP+ INs, whereas a fraction of them is found to co-express reelin and NPY (*Lee et al.,*

*2010*). Using these markers, we find that *Hmx3*; tdTOM+/*Htr3a*-GFP+ INs express reelin and/or NPY, but not VIP, PV or SST. This is in line with previous results showing that *Hmx3+* INs express NPY and not VIP, PV or SST (*Gelman et al., 2009*). Finally, we find that cortical INs from the *Dbx1* + domain express the MGE-enriched markers PV or SST and only rarely co-label with *Htr3a*-GFP + INs. In addition, *Dbx1*-derived cortical INs express the MGE-related TFs SOX6 and LHX6 but not the CGE-enriched TF PROX1. Taken together, our findings thus indicate that *Hmx3+* but not *Dbx1* + cells give rise to a subpopulation of cortical *Htr3a*-GFP+ INs, which share molecular similarities with CGE but not MGE-derived INs. However, given that both *Hmx3+* and *Hmx3- Htr3a*-GFP+ INs express reelin and/or NPY, these classical neurochemical markers are not sufficient to segregate *Hmx3*- and non-*Hmx3*- derived *Htr3a*-GFP+ IN subtypes.

### *Hmx3*-derived *Htr3a*-GFP+ INs display the molecular, morphological and electrophysiological properties of NGCs

Electrophysiological recordings obtained from *Htr3a*-GFP+ INs revealed the existence of many different subtypes of INs (*Lee et al., 2010*). Recently, electrophysiological and morphological characterization of L1 INs combined to single-cell transcriptomics delineated two main types of INs, namely NGCs and SBCs (*Cadwell et al., 2016*). Our findings support this observation and indicate that *Hmx3*-derived *Htr3a*-GFP+ INs exhibit the morphological and electrophysiological signature of NGCs and strongly express *Car4*, a transcript present at high level in NGCs, but not in SBCs. In contrast, *Htr3a*-GFP+ INs in L1 that do not derive from *Hmx3+* cells, have low levels of *Car4* and display the electrophysiological profile of SBCs. These results indicate that *Htr3a*-GFP+ cortical INs in L1 can be subdivided in two major groups characterized by distinct intrinsic properties and that these subgroups are determined by their sites of origin and the differential expression of the TF *Hmx3*. Finally, we show that all *Hmx3*; tdTOM+/*Htr3a*-GFP+ INs analysed in deeper cortical layers also display molecular, morphological and electrophysiological profiles of NGCs, indicating that the *Hmx3*-Cre line labels NGCs across neocortical layers. *In vivo* studies of the canonical cortical microcircuit have mainly relied on the use of the mutually exclusive SST-, PV- and VIP-Cre driver lines (*Cardin et al., 2009*; *Fu et al., 2014*; *Kepecs and Fishell, 2014*; *Pfeffer et al., 2013*; *Pi et al., 2013*; *Pinto and Dan, 2015*; *Sohal et al., 2009*; *Zhang et al., 2014*) but they do not give access to NGCs. These cells are the main source of 'slow' GABA$_B$-receptor mediated inhibition in the neocortex (*Tamás et al., 2003*) and are thought to constitute the core cellular component of a canonical inhibitory circuit in L1 (*Craig and McBain, 2014*). NGCs acts through GABA$_B$-receptors to inhibit the activity of projection neurons and halt ongoing network activity through dendritic calcium channels (*Craig and McBain, 2014*). Long-range interhemispheric inhibition has been shown to be mediated through a GABA$_B$-receptor dependent mechanism and it has been proposed that this process requires the recruitment of L1 cortical INs, possibly of the neurogliaform-type (*Craig and McBain, 2014*; *Palmer et al., 2012*). However, given the diversity of L1 cortical INs (*Cadwell et al., 2016*; *Jiang et al., 2013*) and the lack of molecular tools to specifically target NGCs in vivo, it has so far not been possible to manipulate and interrogate exclusively NGCs in cortical networks. Our findings redefine the *Hmx3*-Cre mice as a valuable tool to specifically investigate the functional contribution of NGCs in the cortical microcircuit motif.

## Materials and methods

**Key resources table**

| Reagent type (species) or resource | Designation | Source or reference | Identifiers | Additional information |
|---|---|---|---|---|
| Strain, strain background (*Mus Musculus*) | Tg(Htr3a-EGFP) DH30Gsat (referred as *Htr3a*-GFP) | GENSAT Consortium | MGI:3846657 | Maintained on a C57Bl/6 background |
| Strain, strain background (*Mus Musculus*) | B6.Cg-Gt(ROSA)26 Sor$^{tm14(CAG-tdTomato)Hze}$/ J (referred as *R26R*-tdTOM$^{fl/fl}$) | The Jackson Laboratory | MGI:104735 | Maintained on a C57Bl/6 background |

*Continued on next page*

*Continued*

| Reagent type (species) or resource | Designation | Source or reference | Identifiers | Additional information |
|---|---|---|---|---|
| Strain, strain background (*Mus Musculus*) | *Tg(Hmx3-icre)1Kess* | provided by Oscar Marin | MGI:5566775 | Maintained on a C57Bl/6 background |
| Strain, strain background (*Mus Musculus*) | *Dbx1tm2(cre)Apie* (referred as *Dbx1-Cre*) | provided by Alessandra Pierani | MGI:3757955 | Maintained on a C57Bl/6 background |
| Antibody | Anti-GFAP, rabbit polyclonal | Abcam, United Kingdom | ab7260 | (1:2000) |
| Antibody | Anti-GFP, rabbit polyclonal | Millipore, Germany | AB3080 | (1:500) |
| Antibody | Anti-GFP, goat polyclonal | Abcam | ab5450 | (1:2000) |
| Antibody | Anti-NeuN (clone A60), mouse monoclonal | Millipore | MAB377 | (1:500) |
| Antibody | Anti-NKX2.1 (H-190), rabbit polyclonal | Santa Cruz Biotechnology, Dallas, TX | sc-13040 | (1:100) |
| Antibody | Anti-NPY, rabbit polyconal | Abcam | ab10980 | (1:500) |
| Antibody | Anti-NR2F2, rabbit polyclonal | Abcam | ab42672 | (1:500) antigen retrieval (Citrate buffer pH 6.0; 85°C; 20 min) |
| Antibody | Anti-PROX1, goat polyclonal | R&D System, Minneapolis, MN | AF2727 | (1:250) |
| Antibody | Anti-PV, mouse monoclonal | Swant, Switzerland | PV235 | (1:2000) |
| Antibody | Anti-Reelin, mouse monoclonal | Abcam | ab78540 | (1:500) |
| Antibody | Anti-S100β, rabbit polyclonal | Abcam | ab41548 | (1:2000) |
| Antibody | Anti-SST, rat monoclonal | Millipore | MAB354 | (1:500) |
| Antibody | Anti-SOX6, rabbit polyclonal | Abcam | ab30455 | (1:500) |
| Antibody | Anti-SOX10 (N-20), goat polyclonal | Santa Cruz Biotechnology | sc-17343 | (1:100) |
| Antibody | Anti-SP8 (C-18), goat polyclonal | Santa Cruz Biotechnology | sc-104661 | (1:50) antigen retrieval (Citrate buffer pH 6.0; 85°C; 20 min) |
| Antibody | Anti-tdTOM, goat polyclonal | Sicgen, Portugal | AB8181-200 | (1:500) |
| Antibody | Anti-VIP, rabbit polyclonal | Abcam | ab22736 | (1:500) ASCF perfusion; 2 hr PFA 4% postfixation |
| Antibody | Donkey anti-rabbit Alexa Fluor488 | Invitrogen, Carlsbad, CA | A21206 | (1:500) |
| Antibody | Donkey anti-goat Alexa Fluor488 | Invitrogen | A11055 | (1:500) |
| Antibody | Donkey anti-goat Alexa Fluor405 | Abcam | ab175664 | (1:500) |

*Continued on next page*

*Continued*

| Reagent type (species) or resource | Designation | Source or reference | Identifiers | Additional information |
|---|---|---|---|---|
| Antibody | Donkey anti-rabbit Alexa Fluor405 | Abcam | ab175651 | (1:500) |
| Antibody | Donkey anti-rabbit Alexa Fluor647 | Invitrogen | A31573 | (1:500) |
| Antibody | Donkey anti-r at Alexa Fluor647 | Invitrogen | A21247 | (1:500) |
| Antibody | Donkey anti-mouse Alexa Fluor647 | Invitrogen | A31571 | (1:500) |
| Antibody | Donkey anti-goat Alexa Fluor647 | Invitrogen | A21447 | (1:500) |
| Antibody | Streptavidin, Alexa Fluor647-conjugated | ThermoFisher/Invitrogen | S21374 | (1:500) |
| Antibody | Anti-Digoxigenin-AP, Fab fragment | Roche, Switzerland | 11082736103 | (1:2000) |
| Recombinant DNA reagent | *Htr3a* plasmid probe | Gift from Dr. B. Emerit | NA | Linearization: HindIII-HF; antisense synthesis: T7; concentration 1 µg |
| Recombinant DNA reagent | *Lhx6* plasmid probe | Gift from Dr. M. Denaxa | NA | Linearization: Not1; antisense synthesis: T3; concentration 1 µg |
| Recombinant DNA reagent | RNAScope Probe-tdTomato-C2 | Affimetrix, Santa Clara, CA | 317041-C2 | |
| Recombinant DNA reagent | RNAScope Probe-EGFP | Affimetrix | 400281 | |
| Recombinant DNA reagent | RNAScope Probe-Car4-C3 | Affimetrix | 468421-C3 | |
| Sequence-based reagent | Genotyping PCR primer for *B6.Cg-Gt(ROSA)26 Sortm14 (CAG-tdTomato)Hze*/J | Microsynth, Switzerland (desalted; 100 µM stock) | WT-F (oIMR9020): AAG GGA GCT GCA GTG GAG TA | https://www2.jax.org/protocolsdb/f?p=116:5:0::NO:5:P5_MASTER_PROTOCOL_ID, P5_JRS_CODE: 29436,007909 |
| Sequence-based reagent | Genotyping PCR primer for *B6.Cg-Gt(ROSA)26 Sortm14 (CAG-tdTomato)Hze*/J | Microsynth (desalted; 100 µM stock) | WT-R (oIMR9021): CCG AAA ATC TGT GGG AAG TC | |
| Sequence-based reagent | Genotyping PCR primer for *B6.Cg-Gt(ROSA)26 Sortm14 (CAG-tdTomato)Hze*/J | Microsynth (desalted; 100 µM stock) | Mut-R (oIMR9103): GGC ATT AAA GCA GCG TAT CC | |
| Sequence-based reagent | Genotyping PCR primer for *B6.Cg-Gt(ROSA)26 Sortm14(CAG-tdTomato)Hze*/J | Microsynth (desalted; 100 µM stock) | Mut-F (oIMR9105): CTG TTC CTG TAC GGC ATG G | |
| Sequence-based reagent | Genotyping PCR primer for Tg(Htr3a-EGFP)DH30Gsat | Microsynth (desalted; 100 µM stock) | Com-F (273): GCA AGA TGT GAC CAA GCC ACC TAT TT | http://www.med.unc.edu/mmrrc/resources/genotyping-protocols/mmrrc-273 |

*Continued on next page*

*Continued*

| Reagent type (species) or resource | Designation | Source or reference | Identifiers | Additional information |
|---|---|---|---|---|
| Sequence-based reagent | Genotyping PCR primer for Tg(Htr3a-EGFP)DH30Gsat | Microsynth (desalted; 100 µM stock) | WT-R: CAG CCC TCA GCC CTT TGA GAC TTA AG | |
| Sequence-based reagent | Genotyping PCR primer for Tg(Htr3a-EGFP)DH30Gsat | Microsynth (desalted; 100 µM stock) | Mut-R: TGA ACT TGT GGC CGT TTA CGT CG | |
| Sequence-based reagent | Genotyping PCR primer for Tg(Hmx3-icre)1Kess | Microsynth (desalted; 100 µM stock) | Mut-F: CTC TGA CAG ATG CCA GGA CA | |
| Sequence-based reagent | Genotyping PCR primer for Tg(Hmx3-icre)1Kess | Microsynth (desalted; 100 µM stock) | Mut-R: TCT CTG CCC AGA GTC ATC CT | |
| Sequence-based reagent | Genotyping PCR primer for Dbx1$^{tm2(cre)Apie}$ | Microsynth (desalted; 100 µM stock) | WT-F (1307): GCA AGG AAA TGT CTC TGG GAC | https://www.infrafrontier.eu/sites/infrafrontier.eu/files/upload/public/pdf/genotype_protocols/EM01924_geno.pdf |
| Sequence-based reagent | Genotyping PCR primer for Dbx1$^{tm2(cre)Apie}$ | Microsynth (desalted; 100 µM stock) | WT-R (1115): GAG GAT GAG GAA ATC ACG GTG | |
| Sequence-based reagent | Genotyping PCR primer for Dbx1$^{tm2(cre)Apie}$ | Microsynth (desalted; 100 µM stock) | Mut-F (cre83): GTC CAA TTT ACT GAC CGT ACA CC | |
| Sequence-based reagent | Genotyping PCR primer for Dbx1$^{tm2(cre)Apie}$ | Microsynth (desalted; 100 µM stock) | Mut-R (cre85): GTT ATT CGG ATC ATC AGC TAC ACC | |
| Commercial assay or kit | VECTASTAIN Elite ABC-HRP Kit | VectorLab, Burlingame, CA | PK-6100 | Manufacturer's protocol |
| Commercial assay or kit | DAB Peroxidase (HRP) Substrate Kit (with Nickel), 3,3'-diaminobenzidine | VectorLab | SK-4100 | Manufacturer's protocol |
| Commercial assay or kit | RNAscope Fluorescence Multiplex Reagent Kit | Advanced Cell Diagnostics, Newark, CA | 320850 | Manufacturer's protocol (fresh frozen tissue) |
| Chemical compound, drug | Nε-(+)-Biotinyl-L-lysine (biocytin) | Sigma Aldrich, Germany | B4261 | Used at 8.1 mM |
| Chemical compound, drug | Fast Red tablets | Kem-En-Tech, Denmark | 4210 | Manufacturer's protocol |
| Chemical compound, drug | Hoechst 33258 | Sigma Aldrich | 23491-45-4 | (1:10000) |
| Chemical compound, drug | Thiopental Inresa 0.5 g | Inresa Arzneimittel GmbH, Germany | | Used at 50 mg/kg |
| Software, algorithm | Microsoft Office 2017 (Excel, Word) | © 2017 Microsoft, Redmond, WA | v.16.9.1 | Manuscript editing |
| Software, algorithm | Adobe Suit CC (Photoshop, Illustrator, Acrobat) | Adobe Systems, San José, CA | v.22.0.1 | Image treatment, figure editing |
| Software, algorithm | GraphPad Prism | GraphPad software, Inc., La Jolla, CA | v.7.0 | Statistics, graph editing |
| Software, algorithm | Fiji | doi:10.1038/nmeth.2019 | v.2.0.0 | Image editing, manual counting |
| Software, algorithm | EndNote X | Thomson Reuters, Canada | v.7.7.1 | Reference editing, bibliography |

*Continued on next page*

*Continued*

| Reagent type (species) or resource | Designation | Source or reference | Identifiers | Additional information |
|---|---|---|---|---|
| Software, algorithm | Neurolucida v.11.02.1 | Microbrightfield, MBF Bioscience, Williston, VT | v.11.02.1 | Neuron reconstruction |
| Software, algorithm | Neurolucida Explorer | Microbrightfield, MBF Bioscience | v.11.02.1 | Morphological reconstruction editing |
| Software, algorithm | MatLab | MathWorks, Natick, MA | | Electrophysiological recordings measurment/editing |
| Software, algorithm | Ephus (MATLAB-based) | doi: 10.3389/fncir.2010.00100 | v. 2.1.0 | Electrophysiological recordings data aquisition |
| Software, algorithm | Clampfit | Molecular Devices, San José, CA | v. 10.1.0.10 | Electrophysiological recordings offline analysis |
| Software, algorithm | R programming language | www.R-project.org | v. 3.4.0 | Statistics |
| Software, algorithm | R package bmrm | doi:10.1038/ncomms 14219 | v. 3.5 | Prediction model |
| Software, algorithm | NLMorpholoy Converter | http://neuronland.org/NLMorphology Converter/NL MorphologyConverter.html | v. 0.8.1 | Morphological reconstruction editing |
| Software, algorithm | R package NeuroAnatomy Toolbox | doi: 10.1016/j.neuron.2016.06.012 | v. 1.8.12.9000 | Morphological reconstruction editing |

## Animals

Animal experiments were approved by the local Geneva animal care committee (GE113/16) and conducted according to international and Swiss guidelines. Mice were housed in the conventional area of the animal facility of the Geneva Medical Center. Water and food were provided *ad libitum* and both temperature (22 ± 2°C) and dark/light cycles (12 hr each) were controlled. Timed-pregnant mice were obtained by overnight mating and the following morning was counted as embryonic day (E) E0.5. *Tg(Htr3a-EGFP)DH30Gsat* mice expressing the enhanced GFP under the control of the *Htr3a* regulatory sequences (*Htr3a*-GFP) were provided by the GENSAT Consortium and maintained on a C57Bl/6 background (*Murthy et al., 2014*). *B6.Cg-Gt(ROSA)26Sor^tm14(CAG-tdTomato)Hze/*J loxP flanked reporter mice (*R26R*-tdTOM^fl/fl) were obtained from Jackson Laboratory. *Htr3a*-GFP mice were crossed with *R26R*-tdTOM^fl/fl mice to obtain *Htr3a*-GFP; *R26R*-tdTOM^fl/fl mice. *Tg(Hmx3-icre) 1Kess* (*Hmx3*-Cre) mice were obtained from Oscar Marin and previously described (*Gelman et al., 2009*). *Dbx1^tm2(cre)Apie* (*Dbx1*-Cre) mice were obtained from Alessandra Pierani and previously described (*Gelman et al., 2011*). Details of the genotyping procedure are given in the Key Resources Table.

## Tissue processing and immunohistochemistry

Pregnant females were euthanized by lethal intraperitoneal (i.p.) injection of pentobarbital (50 mg/kg), embryos were collected by caesarian cut and brains dissected and fixed overnight (O.N.) in cold 4% paraformaldehyde dissolved in 0.1M phosphate buffer (PFA) pH 7.4. For postnatal brains, animals were deeply anesthetized by i.p. injection of pentobarbital and transcardially perfused with 0.9% saline/liquemin followed by cold 4% PFA. Brains were cut on a Vibratome (Leica VT1000S) at 60 μm for immunohistochemistry (IHC) or at 80–100 μm for free-floating *in situ* hybridization (ISH). Sections were kept in a cryoprotective solution at −20°C or processed directly for IHC or ISH as described (*Murthy et al., 2014*). The following primary antibodies were used: rabbit anti-GFAP (1:2000, Abcam), goat anti-GFP (1:2000, Abcam), rabbit anti-GFP (1:500, Millipore), mouse anti-

NeuN (1:500, Millipore), rabbit anti-NKX2.1 (1:100, Santa Cruz Biotechnology), rabbit anti-NPY (1:500, Abcam), rabbit anti-NR2F2 (1:500, Abcam), goat anti-PROX1 (1:250, R and D System), mouse anti-Parvalbumin (PV) (1:2000, Swant), mouse anti-Reelin (1:500, Abcam), rabbit anti-S100β (1:2000, Abcam), rat anti-Somatostatin (SST) (1:500, Millipore), rabbit anti-SOX6 (1:500, Abcam), goat anti-SOX10 (1:100, Santa Cruz Biotechnology), goat anti-SP8 (1:50, Santa Cruz Biotechnology), goat anti-tdTomato (tdTOM) (1:500, Sicgen), rabbit anti-VIP (1:500, Abcam). Secondary goat or donkey Alexa-405,–488, −568 and −647 antibodies (Abcam, Invitrogen) raised against the appropriate species were used at a dilution of 1:500 and sections were counterstained with Hoechst 33258 (1:10000) when no Alexa-405 staining was done. A list of the antibodies is given in the Key Resources Table.

### *In situ* hybridization and RNAscope

Sections were hybridized with the respective DIG-labeled RNA probes as described previously (*Murthy et al., 2014*). The *Htr3a* plasmid probe was linearized with *HindIII-HF* for antisense RNA probe synthesis by T7 polymerase (kind gift from Dr. B. Emerit). The *Lhx6* plasmid probe (*Liodis et al., 2007*) was linearized with Not1 for antisense RNA probe synthesis by T3 polymerase (kind gift from Dr. M. Denaxa). The unbound probe was washed and slices incubated with alkaline phosphatase-conjugated anti-DIG antibody (1:2000, Roche) O.N. at 4°C. Fast Red (Kem-En-Tech) was used as an alkaline phosphatase fluorescent substrate to reveal the hybridized probe. We took advantage of the removal of both GFP and tdTOM endogenous fluorescence due to protocol treatments and revealed them by IHC using green and far-red emitting secondary antibodies, respectively. For illustration purposes, the bound probe (red) and the tdTOM (far red) are shown in blue and red, respectively. For RNAscope experiments, P30 brains were rapidly extracted and fresh frozen. After dehydration and protease treatment, coronal 12 μm-thick brain sections were processed using the RNAscope Multiplex Fluorescent Reagent Kit (Advanced Cell Diagnostics) according to the manufacturer's protocol. Probes targeting mRNAs of the *GFP* and *tdTOM* transgenes and of the endogenous *Car4* gene were designed by Advanced Cell Diagnostics.

## Imaging and quantification of interneuron identity and distribution

Images were acquired using confocal microscopes (Nikon A1R or Axio Imager.Z2 Basis LSM 800) equipped with oil-immersion 40x, 60x and 63x objectives (CFI Plan Fluor 40x/1.3 and CFI Plan Apo VC H 60x/1.4, Nikon or Plan-APO (UV) VIS-IR 40x/1.4 and Plan-Apochromat f/ELYRA 63x/1.4, LSM). For widefield illustrations (*Figure 2—figure supplement 1*), images were taken with Axioscan.Z1 slidescanner (Zeiss), equipped with Plan-Apochromat 10x/0.45 objective (Zeiss). Images were lightly treated (gamma, brightness and and despeckle filter only) for visual purpose with Photoshop CC and manual counts were achieved with Fiji. Data are presented as brain averages calculated from at least three slices at different rostro-caudal levels per brain (except for P5 Dbx1 brain 3). A detailed description of the counts, cells and brains in the different experiments is given in *Supplementary file 1*.

## Electrophysiological recordings and morphological tracing

300 μm-thick coronal brain slices were prepared from 3 to 4 weeks old *Hmx3*; tdTOM+/*Htr3a*-GFP + mice with a vibratome (Leica VT 1000S). In the recording chamber, slices were continuously superfused with ACSF (32°C) containing (in mM): NaCl (119), KCl (2.5), CaCl$_2$ (2.5), MgSO$_4$ (1.3), NaH$_2$PO$_4$ (1.0), NaHCO$_3$ (26.2), and glucose (22), and equilibrated with 95% O$_2$/5% CO$_2$, pH 7.4. Whole-cell recordings were obtained from visually identified *Hmx3*; tdTOM+/*Htr3a*-GFP+ in cortical layers 1–6 and *Hmx3*; tdTOM-/*Htr3a*-GFP+ INs in L1, using an upright microscope (Zeiss Axioskop FS) equipped with differential interference contrast and standard epifluorescence. Borosilicate glass patch pipettes had a resistance of 5–6 MΩ when filled with an internal solution containing (in mM): K gluconate (135), KCl (4), HEPES (10), Phosphocreatine (10), Mg-ATP (4), Na-GTP (0.3), and biocytin (8.1). Current clamp recordings were performed at rest and firing properties were studied by delivering consecutive current pulses, 500 ms duration each, ranging from −20 to +360 pA with a 5 pA increment, every 3 s. Data were acquired using a Multiclamp 700B Amplifier (Molecular Devices), and digitized at 10 kHz (National Instruments), using MATLAB (MathWorks)-based Ephus software (Ephus; The Janelia Farm Research Center). Offline analysis was performed using Clampfit (Version 10.1.0.10, Molecular Devices). Cells were accepted for analysis only if their series resistance was

below 30 MΩ and did not change more than 20% during recordings. Following patch-clamp recordings, slices were incubated in ACSF for 1–2 hr at room temperature, then fixed overnight with 4% PFA / 2% Glutaraldehyde in 0.1 M phosphate buffer. Biocytin-filled recorded cells were revealed with IHC, using streptavidin-Alexa 647 conjugate (1:500, Thermo Fisher Scientific), and confirmed being in L1 and expressing *Hmx3*; tdTOM and/or *Htr3a*-GFP. For detailed morphology, slices were quenched for endogenous peroxidase activity in methanol/0.5% $H_2O_2$, blocked in 0.05 M Tris buffer pH 7.4/0.6% NaCl/0.3% Triton X-100/10% normal horse serum (NHS) and incubated (O.N., 4°C) with avidin-biotin complex (Vectastain Elite ABC-HRP Kit) in 0.1M Tris buffer pH 7.7. 3,3'-diaminobenzidine (DAB) revelation was performed following the manufacturer's protocol (Vectastain DAB Kit; SK-4100). Slices were finally dehydrated in graded series of ethanol/xylene and mounted in Eukitt (Sigma). Morphological reconstructions of biocytin-filled cells were performed with Neurolucida software (v. 11.02.1, MBF Bioscience, Microbrightfield), linked to a microscope (Nikon eclipse 80i) equipped with an oil-immersion 100x objective (Plan Apo VC/1.4, Nikon). Brightfield images of the reconstructed cells were acquired with the same microscope and a 10x objective (Plan Apo/0.45, Nikon). Traces were extracted with Neurolucida Explorer (v.11.02.1, MBF Bioscience, Microbrightfield). 14 *Hmx3*; tdTOM+/*Htr3a*-GFP+ cells (7 in L1, 5 cells in L2/3 and 2 cells in L5) and 3 *Hmx3*; tdTOM-/*Htr3a*-GFP+ INs (3 in L1) from four brains were recovered for morphology. For L1 cells, border artefacts in morphological tracings due to tissue compression were corrected. Traces from 14 *Hmx3*; tdTOM+/*Htr3a*-GFP+ and 3 *Hmx3*; tdTOM-/*Htr3a*-GFP+ INs from four brains were analysed blindly. The membrane resistance (Rm), the membrane resting potential and five properties of the first action potential (AP) at rheobase - i) threshold potential (Vth); ii) AP amplitude (peak); iii) AP latency from current step onset (delay); iv) after-hyperpolarization potential amplitude (AHP) and when present; v) after-depolarization potential amplitude (ADP) - were measured for all recorded cells. Both electrophysiological features and morphological tracings were analyzed blindly and data were attributed back to their corresponding cell. Values for each recorded cell are provided in *Figure 6—source data 2* and *Figure 6—figure Supplement 3—source data 1*.

## Statistical analysis and prediction model

Animals were used regardless of their sex and statistical analysis was done with R programming language and GraphPad Prism. No statistics were used to determine optimal group sample size; however, sample sizes were similar to those used in previous publications from our group and others. Normality of the samples was assessed with D'Agostino-Pearson test and when distribution was not normal, non-parametric tests were applied. Using *bmrm* (v3.3) package for L1-regularized logistic regression model, data were standardized, and a L1-regularized logistic regression model was trained to distinguish between *Htr3a*-GFP+ INs that were *Hmx3*-derived and those which were not. This model assigned a linear weight that reflects the power of each feature in the model logistic regression and computed a probability that a given cell is *Hmx3*-derived in such a way that the misclassification error on the training data was minimized (*Figure 6I–J*). Classification performance of the L1-regularized logistic regression algorithm was assessed by leave-one-out-cross-validation (LOOCV). It consists in training a model on all but one cell, feeding the model with this isolated cell to predict its origin and finally assessing if the prediction is correct. Looping it over all cells, yields a prediction value for each cell, which is used to estimate the generalization error of the classifier. Finally, in order to determine if the prediction made by the logistic regression model improved over the signal contained into each feature taken individually, receiver operating characteristic (ROC) curves were drawn to visualize the sensitivity/specificity ratio for each feature and for the leave-one-out predictions. Areas under the curves (AUC) were analyzed to determine the strongest signals (*Figure 6—figure supplement 2*, *Figure 6—source data 2*).

## Acknowledgements

We thank Oscar Marin and Nicoletta Kessaris for providing the *Hmx3*-Cre mice and Alessandra Pierani for providing the *Dbx1*-Cre mice. This work was supported by a Swiss National Foundation (SNF) Synapsy grant (51NF40-158776) and a SNF (31003A_155896/1) grant for AD.

## Additional information

### Funding

| Funder | Grant reference number | Author |
|---|---|---|
| Schweizerischer Nationalfonds zur Förderung der Wissenschaftlichen Forschung | Synapsy 51NF40-158776; 31003A_153448 and CRSII3_154453 | Anthony Holtmaat |
| International Foundation for Research in Paraplegia | Chair Alain Rossier | Anthony Holtmaat |
| Schweizerischer Nationalfonds zur Förderung der Wissenschaftlichen Forschung | Synapsy 51NF40-158776 and 31003A_155896/1 | Alexandre Dayer |

The funders had no role in study design, data collection and interpretation, or the decision to submit the work for publication.

### Author contributions

Mathieu Niquille, Greta Limoni, Foivos Markopoulos, Data curation, Formal analysis, Investigation, Writing—review and editing; Christelle Cadilhac, Formal analysis, Investigation, Writing—review and editing; Julien Prados, Data curation, Formal analysis, Writing—review and editing; Anthony Holtmaat, Supervision, Funding acquisition, Validation, Methodology, Writing—review and editing; Alexandre Dayer, Conceptualization, Supervision, Funding acquisition, Validation, Investigation, Methodology, Writing—original draft, Administration, Writing—review and editing

### Author ORCIDs

Mathieu Niquille (iD) https://orcid.org/0000-0002-2860-3861
Greta Limoni (iD) http://orcid.org/0000-0002-6808-9510
Foivos Markopoulos (iD) http://orcid.org/0000-0002-5501-9249
Julien Prados (iD) https://orcid.org/0000-0002-8546-241X
Anthony Holtmaat (iD) https://orcid.org/0000-0002-7577-0769
Alexandre Dayer (iD) https://orcid.org/0000-0002-4490-9780

### Ethics

Animal experimentation: Animal experiments were approved by the local Geneva animal care committee and conducted according to international and Swiss guidelines.(GE113/16)

### Decision letter and Author response

Decision letter https://doi.org/10.7554/eLife.32017.033
Author response https://doi.org/10.7554/eLife.32017.034

## Additional files

### Supplementary files

• Supplementary file 1. Detailed description of the technical (brains) and biological (cells) replicates used in the different experiments.
DOI: https://doi.org/10.7554/eLife.32017.030

• Transparent reporting form
DOI: https://doi.org/10.7554/eLife.32017.031

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
