## [Decision Letter]

Thank you for submitting your article "Neurogliaform cortical interneurons expressing the serotonin receptor 3A originate from the preoptic area" for consideration by *eLife*. Your article has been reviewed by three peer reviewers, and the evaluation has been overseen by Joseph Gleeson as the Reviewing Editor and a Senior Editor. The reviewers have opted to remain anonymous.

The reviewers have discussed the reviews with one another and the Reviewing Editor has drafted this decision to help you prepare a revised submission.

Summary:

The paper by Niquille et al. reports that a specific cortical interneuron type called Neurogliaform (NGFC) comes from an embryonic domain in the mouse brain, called Preoptic Area (POA). The authors perform a careful and quite detailed analysis at different development stages using a variety of tools, which include compact transgenic mouse lines, immunohistochemistry, sensitive in situ hybridization, as well as physiology to prove their point. The techniques are appropriate and the data is of high quality. The most important result of the study is the identification of these interneurons as a population of NGCs, a specific class of cortical interneurons. This is a timely study, as researchers are trying to understand cortical interneuron diversity in general, as well as its developmental origin. Among all the other interneuron types that have been reported, NGFC is probably one of the least studied ones, as to date there are no genetic tools to label and target them. This study puts forth that the *Nkx5.1*Cre BAC transgenic mouse line is that missing tool.

This study is a follow-up of the Gelman DM et al., 2009 and 2011 work, showing that a population of cortical interneurons comes from the POA, as opposed to the MGE or CGE. In the 2009 paper and using the same *Nkx5.1*-Cre BAC mouse line, the authors report that the majority of cells seem to belong a more or less single type of interneuron, based on action potential firing and morphological features, which were consistent with previous functional studies on NGFCs. Nevertheless, in that paper, the authors did not name it NGFC and due to the lack of additional makers that have come up since, did not stain for reelin, *Car4*, NR2F2 or Sp8. In addition the paper did not do a detailed characterization of the passive and active electrophysiological properties of the fate-mapped cells. All three reviewers agree that the manuscript adds substantially to the field of cortical interneuron development, but that still requires additional clarifications/support for its conclusions.

Essential revisions:

Figure 1: This study relies heavily on two different BAC transgenic lines – *Nkx5.1*(BAC)-Cre, i.e., *Tg(Hmx3-iCre)1Kess*, and *Htr3a*(BAC)-EGFP, i.e., *Tg(Htr3a-EGFP)DH30Gsat* – that, like many BAC transgenics, are complicated reagents whose respective expression profiles may not always accurately recapitulate their corresponding genetic loci. The use of *Htr3a*(BAC)-EGFP as a proxy for CGE origin is clearly not straightforward in the E14.5 embryo, e.g., there is robust ectopic EGFP expression evident in the cortical plate (Figure 1). The overlap between *Nkx5.1*(BAC)-Cre; Ai14 ("*Nkx5.1*-tdTom") and *Htr3a*(BAC)-EGFP ("*Htr3a-EGFP*") in the E14.5 POA is convincing, but there is also overlap in the ventral CGE (1B), and because *Nkx5.1* is not expressed in POA progenitors but only in migratory precursors it is difficult to assign a precise location to this lineage's origin. Are there *Nkx5.1*-tdTOM+/*Htr3a*(BAC)-EGFP cells that remain in the POA at postnatal ages? If so, it would be difficult to distinguish whether the double positive cells highlighted in Figure 1 give rise to the cortical interneurons analyzed vs. lineages that remain in the POA. Because *Nkx5.1* is not expressed in progenitors, the absence of overlap with *Nkx2.1* (Figure 1) does not prove that these lineages do not arise from an *Nkx2.1*+ territory; in fact, Flames et al. presented evidence (as does Figure 1) that *Nkx2.1* is expressed in the entire POA progenitor pool, even though Lhx6 expression is not maintained in POA lineages, in contrast to those arising from the MGE. Recent fate maps using an *Nkx2.1*-Flpo knock in line (vs. the *Nkx2.1*(BAC)-Cre, whose expression was known to be excluded from the *Nkx6.2* territory) show little if any labeling of layer 1 – this does not prove that layer 1 cells do not arise from the POA, but it puts more weight on the accuracy of the *Nkx5.1*(BAC)-Cre line as a proxy for POA origin. How do the fate maps in this study compare with those shown in Gelman et al. 2009? Has there been any 'drift' in the recombination mediated by this BAC tg over time?

There are tdTOM+/GFP+ cells in all layers of the neocortex, yet the authors have only characterised the electrophysiological properties of cells in layer 1. Are neurons in other layers also NGCs? Do they express *Car4* and have the characteristic morphology of NGCs? A detailed characterization of these neurons throughout all layers is not expected, but it is important to clarify this point.

Quantifications are also needed for panels 1C-E. In relation to the expression of *Nkx2-1*, this transcription factor is downregulated as cells migrate towards the cortex (Nobrega-Pereira et al., 2008). Previous work that has suggested that NGCs derive from both the MGE and the CGE (Tricoire et al., J Neurosci 2010), so it would be important to determine in the tdTOM+/GFP+ cells studied here derive from *Nkx2-1* progenitor cells in the POA. On that note, the authors should correct the statement that *Nkx2-1* is an MGE-specific transcription factor. This is not correct: *Nkx2-1* is also expressed in the ventricular zone of the POA [Flames et al., 2007]). The tdTOM+/GFP+ cells may downregulate *Nkx2-1* by the time they have left the ventricular zone. This is a complicated experiment, because it would probably involve crossing *Nkx2-1*-Flp and *Nkx5.1*-Cre lines with a dual reported, but this is necessary to support the claim "these results indicate that a fraction of *Nkx5.1*+ progenitors in the POA express 5-HT3AR and a pattern of TFs more related to CGE than to MGE-derived INs".

The authors should include morphological reconstructions of the axon of tdTOM+/GFP+ cells to reinforce the notion that these cells are indeed NGCs. The axon of these cells is so characteristic that is almost impossible to miss. Another interesting property of NGCs (at least in hippocampus and piriform cortex) is persistent firing that outlasts the injected current (Sheffield et al., Nature Neurosci 2011; Suzuki et al., Front Cell Neurosci 2014). Is this property also observed in S1 tdTOM+/GFP+ cells?

Additional experiments requested by reviewers but not essential. If not performed, at least address in the text:

Since NGFCs have also been reported in other cortical layers, it would be good if the authors recorded the *Htr3a*-GFP and *Nkx5.1*-tdtomato positive fate-mapped cells in other layers besides layer 1. Although beyond the scope of the paper, it would also be very interesting to check the fate-mapped cells in other brain regions the cells have shown to be present, such as the amygdala. The study would be more complete if the authors morphologically reconstructed some cells from several layers (can be combined with point 1) to show that they fit the pattern observed in other studies.

Perform a limited number of paired-recordings between interneurons and pyramidal cells to show that the *Htr3a*-GFP and *Nkx5.1*-tdtomato positive cells have the very characteristic slow GABAA and GABAB receptor -dependent synaptic output of NGFCs and that *Htr3a*-GFP only cells do not. The reviewer does realize that this is a significant and time-consuming task, but if achieved, it would leave very little room for disputes on whether these cells are NGFCs.

---

## [Author Response]

Essential revisions:Figure 1: This study relies heavily on two different BAC transgenic lines – Nkx5.1(BAC)-Cre, i.e., Tg(Hmx3-iCre)1Kess, and Htr3a(BAC)-EGFP, i.e., Tg(Htr3a-EGFP)DH30Gsat – that, like many BAC transgenics, are complicated reagents whose respective expression profiles may not always accurately recapitulate their corresponding genetic loci. The use of Htr3a(BAC)-EGFP as a proxy for CGE origin is clearly not straightforward in the E14.5 embryo, e.g., there is robust ectopic EGFP expression evident in the cortical plate (Figure 1). The overlap between Nkx5.1(BAC)-Cre; Ai14 ("Nkx5.1-tdTom") and Htr3a(BAC)-EGFP ("Htr3a-EGFP") in the E14.5 POA is convincing, but there is also overlap in the ventral CGE (1B), and because Nkx5.1 is not expressed in POA progenitors but only in migratory precursors it is difficult to assign a precise location to this lineage's origin.

In accordance with previous data from the Fishell and Rudy labs (Lee, S. et al., 2010), we also observe EGFP expression in preplate cortical neurons at early embryonic ages in the *Htr3a*(BAC)-EGFP mouse line. We thus agree that the *Htr3a*(BAC)-EGFP mouse line is not a straightforward proxy for CGE-derived cortical interneurons (INs) at embryonic time-points. However, at postnatal ages, the combination of *Htr3a*(BAC)-EGFP and MGE-derived interneuron driver lines has been shown to be a useful tool to track 100% of cortical INs (Lee et al., 2010). Because *Nkx5.1* is not expressed in POA progenitors and only in migratory precursors, we agree that it is currently difficult to establish that *Nkx5.1*-derived INs originate from progenitors residing in the POA ventricular zone (VZ). We have thus modified the manuscript accordingly and removed the term “progenitor” throughout the manuscript. We now indicate that *Nkx5.1*-derived INs are observed in the embryonic POA region but do not make the claim that they derive from progenitors generated in the POA VZ. Additional tools would be required to answer the specific question of their mitotic progenitor origin (see our fourth response).

Are there Nkx5.1-tdTOM+/Htr3a(BAC)-EGFP cells that remain in the POA at postnatal ages? If so, it would be difficult to distinguish whether the double positive cells highlighted in Figure 1 give rise to the cortical interneurons analyzed vs. lineages that remain in the POA.

We have investigated postnatal ages, and observed that in contrast to the embryonic POA, only very few *Nkx5.1*; tdTOM+ / *Htr3a-*GFP+ cells remained postnatally in subpallial regions corresponding to embryonic POA region, such as the preoptic nuclei (PoN). This observation supports the hypothesis that embryonic double-labelled *Nkx5.1*; tdTOM+ / *Htr3a-*GFP+ cells exit the embryonic POA to populate other brain regions such as cortex. These new data are now provided in Figure 2—figure supplement 2 and are described in the new Results section.

Because Nkx5.1 is not expressed in progenitors, the absence of overlap with Nkx2.1 (Figure 1) does not prove that these lineages do not arise from an Nkx2.1+ territory; in fact, Flames et al. presented evidence (as does Figure 1) that Nkx2.1 is expressed in the entire POA progenitor pool, even though Lhx6 expression is not maintained in POA lineages, in contrast to those arising from the MGE. Recent fate maps using an Nkx2.1-Flpo knock in line (vs. the Nkx2.1(BAC)-Cre, whose expression was known to be excluded from the Nkx6.2 territory) show little if any labeling of layer 1 – this does not prove that layer 1 cells do not arise from the POA, but it puts more weight on the accuracy of the Nkx5.1(BAC)-Cre line as a proxy for POA origin. How do the fate maps in this study compare with those shown in Gelman et al. 2009? Has there been any 'drift' in the recombination mediated by this BAC tg over time?

We agree with these interesting comments, which are also related to the sixth point below. It is true that the progenitors in the VZ of the POA region express high levels of *Nkx2.1* as shown in Flames et al. work (2007). Our new quantification at E14.5 in POA, indicates that only a small fraction (about 15%) of *Nkx5.1*; tdTOM+ / *Htr3a-*GFP+ cells co-express NKX2.1. As suggested by reviewers, it is thus possible that postmitotic cells from the *Nkx5.1* lineage downregulate *Nkx2.1* after exiting the VZ of the POA. However, it should be noted that the*Nkx2.1*-Flpo knock-in line does not appear to label cells in layer 1 (see Figure S1 in He et al., 2016), indicating that *Nkx5.1*-derived NGCs in layer 1 may derive from progenitors that do not belong to the *Nkx2.1* lineage.

To comment on this point, we have now added in the Discussion the following new paragraph with new references:

“In the embryonic POA, we observe that only a small fraction of *Nkx5.1*; tdTOM+ / *Htr3a*-GFP+ cells express the TF NKX2.1, which has been shown to be strongly expressed in the ventricular zone of the POA (Flames et al., 2007). […] Overall, further work needs to be done to clarify the precise origin of mitotic progenitors giving rise to the pool of Nkx5.1; tdTOM+ / Htr3a-GFP+ cells observed in the embryonic POA.”

Finally,although a drift in recombination of the BAC trangenics over time is always possible, the pattern of recombination we have observed appears quite similar to the Gelman et al. publication.

There are tdTOM+/GFP+ cells in all layers of the neocortex, yet the authors have only characterised the electrophysiological properties of cells in layer 1. Are neurons in other layers also NGCs? Do they express Car4 and have the characteristic morphology of NGCs? A detailed characterization of these neurons throughout all layers is not expected, but it is important to clarify this point.

We analysed *Nkx5.1*; tdTOM+ / *Htr3a-*GFP+ INs in deeper cortical layers to determine whether they also displayed NGC characteristics. As for layer 1 cells, *Car4* expression was significantly higher in *Nkx5.1*; tdTOM+ / *Htr3a*-GFP+ INs residing in layer 2-6 as compared to *Htr3a*-GFP+ INs negative for tdTOM (Figure 6—figure supplement 3). Morphological reconstructions of *Nkx5.1*; tdTOM+ / *Htr3a*-GFP+ INs in layers 2-6 revealed that all cells had the characteristic morphology of NGCs with dense and fine axonal ramifications (Figure 6—figure supplement 4). Furthermore, electrophysiological recordings of *Nkx5.1*; tdTOM+ / *Htr3a-*GFP+ INs in cortical layers 2-6 indicated that they displayed a more pronounced tendency to late-spiking compared to layer 1 cells, an electrophysiological property which is a typical of NGCs (Figure 6—figure supplement 3). These new molecular, morphological and electrophysiological data comparing layer 1 and layer 2-6 *Nkx5.1*; tdTOM+ / *Htr3a-*GFP+ INs are available in Figure 6—figure supplement 3, 4 and in the Figure 6—figure supplement 3—source data 1 and 2.

Quantifications are also needed for panels 1C-E.

These quantifications are now provided in the main Figure 1 and in the Results section. Detailed cells counts are available in the Supplementary file 1 and Figure 1—source data 1.

In relation to the expression of Nkx2-1, this transcription factor is downregulated as cells migrate towards the cortex (Nobrega-Pereira et al., 2008). Previous work that has suggested that NGCs derive from both the MGE and the CGE (Tricoire et al., J Neurosci 2010), so it would be important to determine in the tdTOM+/GFP+ cells studied here derive from Nkx2-1 progenitor cells in the POA. On that note, the authors should correct the statement that Nkx2-1 is an MGE-specific transcription factor. This is not correct: Nkx2-1 is also expressed in the ventricular zone of the POA [Flames et al., 2007]). The tdTOM+/GFP+ cells may downregulate Nkx2-1 by the time they have left the ventricular zone. This is a complicated experiment, because it would probably involve crossing Nkx2-1-Flp and Nkx5.1-Cre lines with a dual reported, but this is necessary to support the claim "these results indicate that a fraction of Nkx5.1+ progenitors in the POA express 5-HT3AR and a pattern of TFs more related to CGE than to MGE-derived INs".

We fully agree that further work needs to be done to clarify the precise origin of mitotic progenitors giving rise to *Nkx5.1*; tdTOM+ / *Htr3a-*GFP+ cells observed in the embryonic POA. As correctly mentioned by the reviewers this specific point can only be resolved by a complicated experiment involving additional transgenic mouse lines, including the Nkx2.1-Flpo knock-in line and a dual reporter mouse line. As indicated in our answer to point 1, we have modified the manuscript and now indicate that Nkx5.1-derived INs are observed in the embryonic POA region but do not make the claim that they derive from progenitors generated in the POA ventricular zone. In addition, we have added a new paragraph in the discussion to address this point (see our answer to point 3). Finally, have removed the statement that Nkx2.1 is a MGE-specific transcription factor.

The authors should include morphological reconstructions of the axon of tdTOM+/GFP+ cells to reinforce the notion that these cells are indeed NGCs. The axon of these cells is so characteristic that is almost impossible to miss.

We now provide morphological reconstructions of *Nkx5.1*; tdTOM+ / *Htr3a-*GFP+ INs in layers 1-6. All reconstructed cells displayed the characteristic morphology of elongated NGCs with dense and fine axonal ramifications (Figure 6, Figure 6—figure supplement 1, Figure 6—figure supplement 4).

Another interesting property of NGCs (at least in hippocampus and piriform cortex) is persistent firing that outlasts the injected current (Sheffield et al., Nature Neurosci 2011; Suzuki et al., Front Cell Neurosci 2014). Is this property also observed in S1 tdTOM+/GFP+ cells?

Although the physiological importance of persistent firing is not well understood, we agree that it is interesting to explore whether this property described for NGCs of the hippocampus and the piriform cortex also pertains to NGCs of the neocortex. In our first set of recordings to study the firing properties of *Nkx5.1*-derived NGCs in barrel cortex layer 1 (BC L1), we used protocols of 0.5 s-long subsequent depolarizing current steps with 20 pA increments (similarly to Sheffield et al., Nat Neurosci., 2011). This protocol did not trigger persistent firing in any of the 21 *Nkx5.1*-derived NGCs included in the analysis of our first submission. To verify that this was not merely due to our experimental conditions, we performed additional recordings from 6 *Nkx5.1*-derived NGCs in L1-3, using two different protocols for somatic stimulation described by Suzuki et al., (Front Cell Neurosci,2014) (1 s-long depolarizing current steps (0.5-1.5 nA) at 0.5 Hz, and 2 ms-long depolarizing current steps (1.0-1.5 nA) at 20 Hz). Persistent firing occurred only in 1 out of 6 recorded *Nkx5.1*-derived NGCs (17%), in the second trial of the 1 s-long depolarizing current steps protocol. It lasted for 6.5 s after the 20^th^ depolarizing current step and it did not reoccur in the following three trials of stimulation that took place with 2 min intervals. Our values for frequency of occurrence and duration (17% and 6.5 s respectively) were much lower as compared to the ones reported for NGCs by Suzuki et al., (Front Cell Neurosci, 2014). As reported in the above-mentioned studies, the range of temperature as well as the concentrations of K^+^ and Ca^2+^ in the recording ACSF are critical for persistent firing occurrence and duration. Most probably, the lower temperatures (30-32°C vs. 33-35°C) and the lower K^+^ concentration (2.5 mM vs. 3.0 mM) in our experiments account for the lower frequency of occurrence, whereas the higher Ca^2+^ concentration (2.5 mM vs. 2.0 mM) accounts for the shorter duration of persistent firing. Hence, we conclude that persistent firing can also be induced in *Nkx5.1*-derived NGCs in BC superficial layers, but its occurrence and duration depends strongly on the experimental conditions applied. It will be interesting to further investigate which conditions would allow a higher incidence and duration of persistent firing in these cells. However, we feel that this is beyond the scope of the present study.

Additional experiments requested by reviewers but not essential. If not performed, at least address in the text:Since NGFCs have also been reported in other cortical layers, it would be good if the authors recorded the Htr3a-GFP and Nkx5.1-tdtomato positive fate-mapped cells in other layers besides layer 1. Although beyond the scope of the paper, it would also be very interesting to check the fate-mapped cells in other brain regions the cells have shown to be present, such as the amygdala. The study would be more complete if the authors morphologically reconstructed some cells from several layers (can be combined with point 1) to show that they fit the pattern observed in other studies.

As indicated in our response to point 4, we analysed *Nkx5.1*; tdTOM+ / *Htr3a-*GFP+ INs in deeper cortical layers to determine whether they also displayed NGC characteristics. As for layer 1 cells, *Car4* expression was significantly increased in *Nkx5.1*; tdTOM+ / *Htr3a*-GFP+ INs residing in layer 2-6 as compared to *Htr3a*-GFP+ INs negative for tdTOM (Figure 6—figure supplement 3). Morphological reconstructions of *Nkx5.1*; tdTOM+ / *Htr3a*-GFP+ IN residing in cortical layers 2-6 revealed that all cells had the characteristic morphology of NGCs with dense and fine axonal ramifications (Figure 6—figure supplement 4). Furthermore, recordings of *Nkx5.1*; tdTOM+ / *Htr3a-*GFP+ INs in cortical layers 2-6 indicated that they displayed more pronounced NGC electrophysiological profiles such as late-spiking properties compared to layer 1 cells(Figure 6—figure supplement 3).

Finally, we have observed *Nkx5.1*; tdTOM+ / *Htr3a*-GFP+ fate-mapped cells in a variety of brain regions including the amygdala and hippocampus. These cells are now shown in illustrative rostro-caudal sections in a new Figure 2—figure supplement 1

Perform a limited number of paired-recordings between interneurons and pyramidal cells to show that the Htr3a-GFP and Nkx5.1-tdtomato positive cells have the very characteristic slow GABAA and GABAB receptor -dependent synaptic output of NGFCs and that Htr3a-GFP only cells do not. The reviewer does realize that this is a significant and time-consuming task, but if achieved, it would leave very little room for disputes on whether these cells are NGFCs.

Given that new molecular, morphological and electrophysiological recordings performed in other cortical layers confirm that *Nkx5.1*; tdTOM+ / *Htr3a*-GFP+ have characteristic features of NGCs, this specific point was not addressed.